# Multi-domain interaction mediated strength-building in human α-actinin dimers unveiled by direct single-molecule quantification

Yuhang Zhang[1,6], Jingyi Du[2,6], Xian Liu[2], Fei Shang[1], Yunxin Deng[3], Jiaqing Ye[1], Yukai Wang[1], Jie Yan[3,4,5], Hu Chen[1]✉, Miao Yu[2]✉ & Shimin Le[1]✉

α-Actinins play crucial roles in cytoskeletal mechanobiology by acting as force-bearing structural modules that orchestrate and sustain the cytoskeletal framework, serving as pivotal hubs for diverse mechanosensing proteins. The mechanical stability of α-actinin dimer, a determinant of its functional state, remains largely unexplored. Here, we directly quantify the force-dependent lifetimes of homo- and hetero-dimers of human α-actinins, revealing an ultra-high mechanical stability of the dimers associated with >100 seconds lifetime within 40 pN forces under shear-stretching geometry. Intriguingly, we uncover that the strong dimer stability is arisen from much weaker sub-domain pair interactions, suggesting the existence of distinct dimerized functional states of the dimer, spanning a spectrum of mechanical stability, with the spectrin repeats (SRs) in folded or unfolded conformation. In essence, our study supports a potent mechanism for building strength in biomolecular dimers through weak, multiple sub-domain interactions, and illuminates multifaceted roles of α-actinin dimers in cytoskeletal mechanics and mechanotransduction.

In cells, the actomyosin cytoskeleton comprises actin filaments, actin-binding proteins that orchestrate the assembly of these filaments into various structures, and myosin fibers that induce dynamic filament contractions. This actomyosin cytoskeleton is intricately connected to the cell membrane, nuclear membrane, as well as other cytoskeletal elements, such as intermediate filaments and microtubules, through sets of adaptor proteins[1–4]. Through actomyosin contractions and external mechanical perturbations, dynamic tension is generated and propagated along the filaments, resulting in supramolecular networks that are both highly dynamic and remarkably robust. These networks enable the mechanosensing and mechanotransduction of cells, governing various processes including cell–matrix and cell–cell adhesions, cell migration, cell differentiation, growth, tissue repair, and embryonic development[1–3].

A pivotal family of actin-binding proteins within this robust cytoskeletal system is the α-actinins, which are ubiquitously conserved across animals[5–12]. An α-actinin monomer comprises an N-terminal actin-binding domain (ABD) consisting of two consecutive calponin homology (CH) domains[13], a central rod domain composed of four spectrin-like repeats (SR)[14,15], and a C-terminal calmodulin-like domain (CaMD) formed by four EF hands (EF12 and EF34)[16,17]. Each SR adopts a characteristic bundle of three antiparallel α-helices with a length of 106–122 amino acids[15]. Four isoforms of mammalian α-actinins have been genetically and biochemically characterized, classified into two groups: (1) non-muscle isoforms, including α-actinin 1 and 4, which are present in all cells, and (2) sarcomeric-specific isoforms, including α-actinin 2 and 3, particularly crucial in anchoring actin filaments to the muscle Z-line[7,8,18].

[1]Department of Physics, Research Institute for Biomimetics and Soft Matter, Fujian Provincial Key Lab for Soft Functional Materials Research, Xiamen University, Xiamen 361000, China. [2]Department of Biochemistry and Department of Orthopaedic Surgery of the Second Affiliated Hospital, Zhejiang University School of Medicine, Hangzhou 310058, China. [3]Mechanobiology Institute, National University of Singapore, Singapore 117411, Singapore. [4]Department of Physics, National University of Singapore, Singapore 117542, Singapore. [5]Joint School of National University of Singapore and Tianjin University, International Campus of Tianjin University, Fuzhou 350207, China. [6]These authors contributed equally: Yuhang Zhang, Jingyi Du. ✉e-mail: chenhu@xmu.edu.cn; yu.miao@xmu.edu.cn; leshimin@xmu.edu.cn

It is widely acknowledged that $\alpha$-actinins primarily function as dimers in vivo. Normally, $\alpha$-actinins exist as anti-parallel homo-dimers, while anti-parallel hetero-dimers of $\alpha$-actinin 1 and 4[19], or $\alpha$-actinin 2 and 3[20], may be present in specific cell lines. Detailed crystal structures reveal that the two anti-parallel rods pair and twist about 90° along the long axis of the $\alpha$-actinin dimer, resulting in an ABD at each end of the dimer, facilitating the cross-linking of actin filaments[7,15,21]. This structural arrangement is connected to their capability to cross-link actin filaments primarily in an anti-parallel manner, where the anti-parallel dimerized tandem spectrin repeats are stretched under shear–force geometry. In addition, the dimer under unzip-force stretching geometry may also physiologically relevant since the C-terminus EF hands of the molecule are known to bind to various mechanosensitive proteins, such as palladin[22], titin[23]. While there is consensus that $\alpha$-actinins play a crucial role in organizing and maintaining the cytoskeletal framework as force-bearing structural modules and act as signaling hubs for various mechanosensing proteins, an essential determinant of $\alpha$-actinins' functional state—namely, the mechanical stability of the $\alpha$-actinin dimers—remains poorly understood.

Dynamic in vivo tension on actin cytoskeleton-associated mechanotransmission pathways is generated primarily due to acto-myosin contractions and external mechanical perturbations. The physiological force range on individual mechanosensitive proteins, such as vinculin, talin, $\alpha$-catenin, integrins, E-cadherins, etc., have been estimated to be a few pico-Newton (pN) to several tens of pN by fluorescence resonance energy transfer (FRET) experiments utilizing fluorescence tension sensors inserted in or linked to the target mechanosensitive proteins[24–38]. The pN scale forces are consistent with the forces that myosin motors could generate[39,40]. The physiological lifetime of the force-transmission pathways could also be estimated by fluorescence recovery after photobleaching (FRAP) experiments. The lifetime of classic mechanosensitive proteins at focal adhesions and adherens junctions has been estimated to be a few seconds to a few minutes[24,31,41]. The physiological force-loading rates could then be roughly estimated on the order of $pN\,s^{-1}$. Consistently, the force loading rate on integrin has recently been estimated to be ranged from 0.5 to 4 $pN\,s^{-1}$, using DNA-based force-loading sensors[42,43]. The forces of a few pN and lifetime of seconds to minutes have been demonstrated to be sufficient to mechanically modulate the signaling pathways, such as talin–vinculin mediated focal adhesion[44], formin-mediated F-actin polymerization[45,46], GAIN domain dissociation of adhesion-GPCR, etc.[47,48].

The physiological force and lifetime depend on the force-dependent lifetime of the protein–protein interfaces (PPI) of the force-bearing proteins on the pathways. Recently, force-dependent lifetime of several critical PPIs has been quantified. For instance, the PPI between the vinculin tail domain and F-actin withstands forces of ~8 pN for ~10 s[49], while the PPI between the talin ABS3 (actin-binding site 3) and F-actin withstands forces of ~10 pN for tens of seconds[50]. The PPI between vinculin and its VBS on talin/$\alpha$-catenin withstands forces of ~10 pN for hundreds of seconds[51]. The PPI of filamin homo-dimer withstands forces of ~10 pN for a few seconds[52]. However, the force-dependent lifetime of the PPI of $\alpha$-actinin dimer, the primary organizer of the actin cytoskeleton, has not been quantified yet.

Here, utilizing magnetic tweezers[53,54], we directly investigated the force-induced rupture of both homo- and hetero-dimers of human $\alpha$-actinins at the single-molecule level and quantified their force-dependent lifetimes. Our findings reveal that these dimers exhibit ultra-high mechanical stability, boasting lifetimes > 100 s under shear-stretching geometry at forces within 40 pN. This stands in stark contrast to their frail mechanical stability under unzip-stretching geometry, marked by lifetimes in the range of a few seconds at forces above 4 pN. Furthermore, by directly assessing the mechanical stability of dimer formed by sub-domain pairs, we demonstrate that the strong mechanical stability of the entire dimer originates from its inherently weaker sub-domain pairs. We propose a potent mechanism for building strength in biomolecular dimers through weak, multiple sub-domain interactions. Our results also suggest the potential existence of various dimerized functional states of the dimer, characterized by a spectrum of mechanical stability, ranging from weak to strong, thereby enabling mechanical switches in signaling proteins. We extend our observations to the ancestral $\alpha$-actinin, specifically the *Entamoeba histolytica* $\alpha$-actinin-2, which also employs a similar strength building mechanism, implying an evolutionarily conserved strategy governing the mechanical regulation of $\alpha$-actinins. Altogether, these results enrich our understanding of the physical mechanism behind the mechanical stability and the diverse force-dependent structural states of $\alpha$-actinin dimers, thereby illuminating their multifaceted roles in cytoskeletal mechanics and mechanotransduction.

## Results

### High mechanical stability of $\alpha$-actinin 1 homo-dimer under shear–force stretching geometry

Since $\alpha$-actinins form dimers solely via the anti-parallel dimerized SR rods, the mechanical stability of the $\alpha$-actinin dimers equals to the mechanical stability of the anti-parallel SR rod dimers, i.e., the anti-parallel pair of SR1–SR2–SR3–SR4. We designed a single-molecule platform that enables direct mechanical manipulation of individual anti-parallel SR rod dimers of $\alpha$-actinins (Fig. 1a). For each $\alpha$-actinin isoform, a chimeric protein was constructed, consisting of (from the N-terminus to C-terminus) a SpyTag, SR1–SR2–SR3–SR4, a long flexible linker, a biotinylated AviTag, and another SR1–SR2–SR3–SR4 (Fig. 1a, see Methods section: protein constructs list, plasmids cloning and protein expression)[41,51,55–58]. The chimeric protein was specifically tethered between a SpyCatcher-coated coverslip surface and a strep-tavidin/neutravidin-coated superparamagnetic bead via the SpyTag and biotinylated AviTag in the protein, respectively (Fig. S1). Well-controlled external force using magnetic tweezers was applied (see Methods section: single-protein manipulation and analysis, surface preparation, protein tether formation, bead-height change in the magnetic-tweezer setup). The dimer was stretched via the SpyTag at the N-terminus of the first SR rod and the biotinylated AviTag at the N-terminus of the second SR rod, resulting in an anti-parallel SR rod dimer stretched under shear–force geometry, with the long flexible linker looped within the dimer (Fig. 1a). Force-induced rupture of the dimer causes the releasing of the looped long flexible linker, leading to a corresponding force-dependent step-wise bead height jump. In addition to the dimer rupture, unfolding of the SR domains also causes characteristic force-dependent step-wise bead height jumps. The force-dependent step sizes of both dimer rupture and domain unfolding can be theoretically calculated based on polymer physics models[4] (Fig. S3, see Methods section: theoretical calculation of the force-dependent transition step size).

Figure 1b shows representative time–bead height curves of the $\alpha$-actinin 1 rod homo-dimer under a shear–force stretching geometry, recorded during two cycles of force-loading experiments. The force was first kept at ~1 pN (cyan color) and then linear increased to ~80 pN with a force loading rate of 5 $pN\,s^{-1}$ (red color), during which the rupture of the dimer and concurrent unfolding of the domains were observed (red arrows). Then the force was decreased to ~8 pN (with a loading rate of −5 $pN\,s^{-1}$), and switched to linearly slow force-decreasing scan with a loading rate of −0.1 $pN\,s^{-1}$ (blue color) to observe the refolding of the domains and re-dimerization of the dimer. The ruptured and unfolded molecule was able to be refolded and re-dimerized at lower forces (e.g., refold at <6 pN, as indicated by the four gray arrows, and re-dimerizes at ~3 pN forces, as indicated by the dark gray arrow, respectively. Fig. 1b, c–e, right panel), creating a new dimer for next force-increase scan cycle. By repeating multiple cycles of force-loading scans on multiple molecules, we obtained the force–bead height curves of the dimer, the rupture force–stepsize

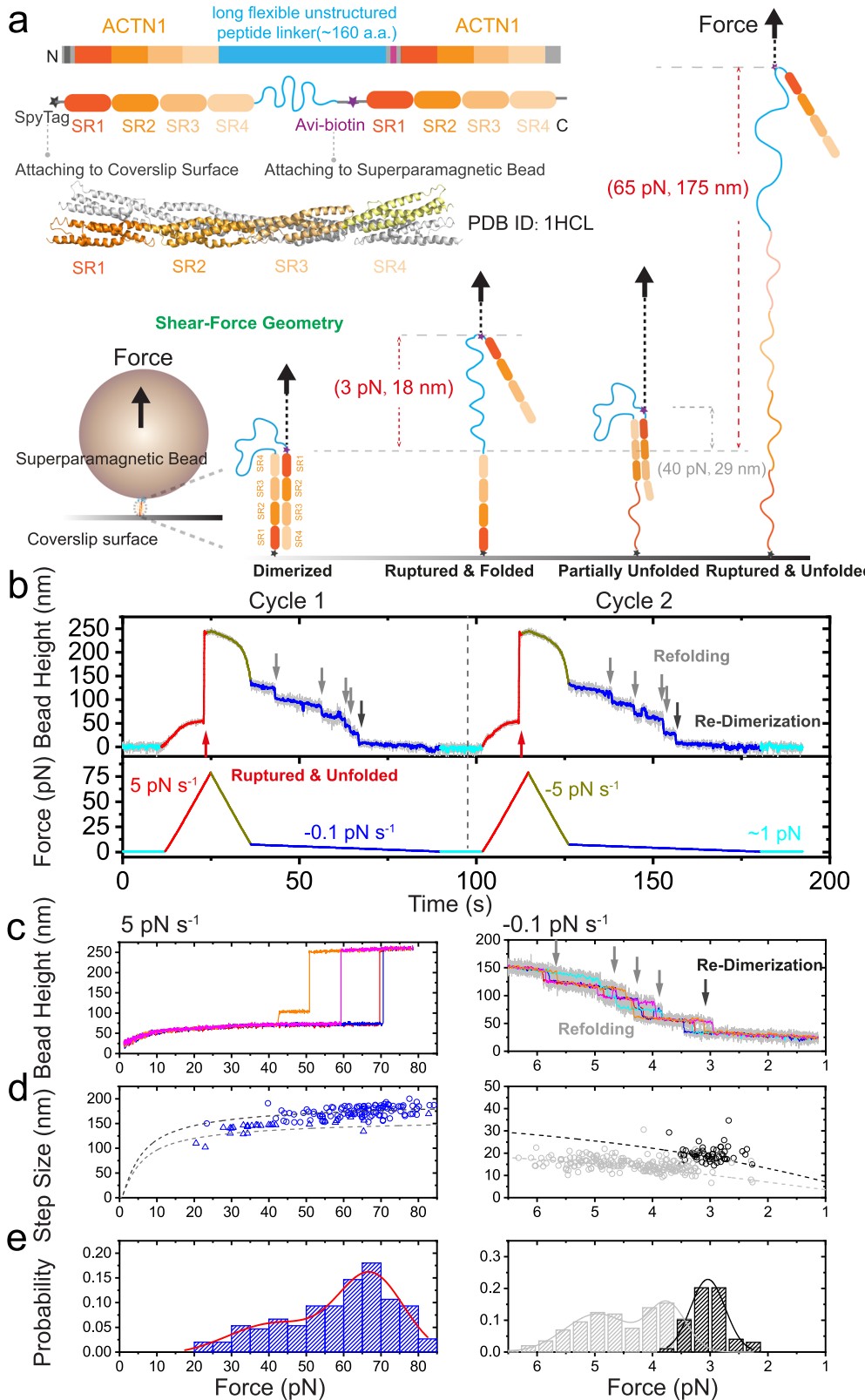

graph of the dimer and the rupture force distributions at 5 pN s$^{-1}$ (Fig. 1c–e, left panel).

Figure 1c shows the representative force–bead height curves of the α-actinin 1 rod homo-dimer under a shear–force stretching geometry during force-increase and force-decrease scans. In line with the theoretically predicted force-dependent rupture step size of the dimer (Fig. S3, see Methods section: theoretical calculation of the

force-dependent transition step size), a large step-wise bead-height jump, exceeding 160 nm, was observed at forces > 60 pN in the majority of the force-increase scans (Fig. 1c, left panel), indicates the rupture of the dimer and concurrent unfolding of the SR1–SR2–SR3–SR4 under stretching. It is because that in the monomer form, the SR3–SR4 domains and SR1–SR2 domains unfold at forces of ~18 pN and ~27 pN, respectively[41], which is much less than 60 pN.

**Fig. 1 | High mechanical stability of human α-actinin 1 homo-dimer under shear–force geometry. a** The illustration of the single-molecule construct and experimental design of the human α-actinin 1 homo-dimer under shear–force geometry. Rupture of the dimer releases the long looped flexible linker with/ without concurrent domain unfolding, leading to large step-wise bead height jumps. The expected step-sizes at example forces are provided on the panel within brackets. **b** The illustration and representative cycles of linear force loading experiments. The force was first held at -1 pN, and then was increased to -80 pN (5 pN s⁻¹) to observe the dimer rupture and domains unfolding (red arrows). The force was then fast decreased to -8 pN (with −5 pN s⁻¹) and decreased to -1 pN (−0.1 pN s⁻¹) to observe four SR refolding (gray arrows) and dimer re-formation (dark gray arrows). **c** The representative force–bead height curves of the dimer during force-increase (5 pN s⁻¹, left panel) and force-decrease (−0.1 pN s⁻¹, right panel) scans. Gray arrows indicate the domains refolding, and dark gray arrow indicates the re-dimerization event. Light gray and colored lines indicate the raw data and its 10-point FFT (Fast Fourier Transform) smoothing, respectively. **d** The force–step size graph of the dimer during force-increase and force-decrease scans. Left panel: the theoretical force-dependent step-sizes assuming the concurrent rupturing and unfolding of all four SR domains (dark gray dashed line,) or with three SR domains (gray dashed line), respectively. Rupture events: N = 150. Right panel: The theoretical force-dependent stepsizes of the dimer formed by fully folded SR rods (dark gray dashed line) and individual SR domains (gray dashed line), respectively. Re-dimerization events: N = 55, refolding events: N = 200. **e** Left panel: the normalized rupture forces distribution. Right panel: the normalized re-dimerization forces distribution (dark gray) and refolding force distribution (gray). The lines are gaussian fitting curves of the distributions, with the peak values (from left to right) as 41.6 ± 7.4 pN, 67.4 ± 2.5 pN, 4.95 ± 0.07 pN, 3.73 ± 0.04 pN and 3.04 ± 0.03 pN, respectively. Source data are provided as a Source Data file.

Once the dimer ruptures at >60 pN, the four SR domains in the resulting monomer under stretching at this force unfold immediately, leading to a single step signal containing rupturing and unfolding events within our experimental time resolution. In some scans, a partial rupture of the dimer, likely involving the SR1:SR4 pair, preceded the complete dimer rupture. This resulted in an unfolding step of one of the sub-domains, followed by the rupture of the dimer and simultaneous unfolding of the remaining domains (orange curve in Fig. 1c, left panel). In addition, occasional instances were noted where dimer rupture occurred at forces lower than 30 pN. This led to the rupture of the dimer along with the unfolding of SR3–SR4, followed by the unfolding of SR1–SR2 (as depicted by the cyan curve in supplementary Fig. S4).

The left panel of Fig. 1d, e shows the resulting force–step size graph and the normalized rupture force distribution of the dimer. The major rupture force peaked at 67 ± 3 pN indicates high mechanical stability of α-actinin 1 dimer under shearing stretching geometry. A minor force peak also emerged at 42 ± 7 pN, suggesting the potential existence of one or more mechanically weaker dimerized states. Interestingly, partial rupture of the dimer, presumably the rupture of the SR1:SR4 pair, indicated by a bead height jump with a stepsize of a domain unfolding, was often observed prior the full rupture of the dimer. These results suggest that the mechanical rupture pathway of the dimer may be initiated with partial rupture of the SR1:SR4 pair.

### Force-dependent lifetime of α-actinin 1 Homo-dimer under shear–force Stretching Geometry

To gain further insight into the mechanical stability of the actinin dimer, we proceeded to directly quantify the force-dependent lifetime of the dimer, by implementing a force–jump-cycle procedure (Fig. 2a). In each cycle, we first held the molecule at a low force of -1 pN where the molecule was in a fully dimerized state (cyan color). We then jumped the force to a target value (e.g., 45 pN in Fig. 2a), and maintained the target force until the rupture event associated with a large step jump was observed, indicated by a red arrow. Subsequently, we fast decreased the force to -7 pN and linearly decreased to -1 pN with a slow loading rate of −-0.1 pN s⁻¹ to observe the four domain refolding steps (gray arrows) and one re-dimerization step (dark gray arrow). This process ensures re-creating a fully dimerized molecule prior the next force–jump-cycle.

Figure 2b shows the representative time–bead height curves at two target forces obtained from multiple force–jump cycles. The dwell time (Δt) of the molecule prior its rupture was recorded for each cycle (Fig. 2a), and the lifetime of the molecule was obtained by fitting to the time-dependent rupture probability based on the dwell time distribution using bootstrap analysis (Fig. S6−8, and see Methods section: effects of the flexible peptide chain linker, bootstrap analysis, Bell model analysis). Consistent with observations from force-increase scan experiments, partial rupture of the dimer—presumably the rupture of the SR1:SR4 pair—indicated by a bead height jump corresponding to domain unfolding, was often observed before the complete rupture of the dimer. Additionally, the results suggest that a partial dimer, likely the SR2–SR3–SR4:SR1–SR2–SR3 configuration, also exhibits considerable mechanical stability even under physiological force levels.

Figure 2c shows the resulting force-dependent lifetime of the α-actinin 1 homo-dimer under shear–force stretching geometry. Clearly, the dimer has an ultra-high mechanical stability associated with >100 s lifetime within 40 pN under shear-stretching geometry, sufficient to support its role as force-bearing structural module and force-transmission in the cytoskeleton. In addition, consistent with the minor rupture force peak at forces of -30 pN (Fig. 1e), occasional molecules with shorter force-dependent lifetime was observed (magenta and orange colors, Fig. 2c), suggesting the existence of multiple mechanically weaker sub-groups of the dimer.

### Weak mechanical stability of α-actinin 1 homo-dimer under unzip-force stretching geometry

While it is in general believed that the dimer under shear–force stretching geometry is the primary functional state in vivo, the dimer under unzip-force stretching geometry may also be physiologically relevant since the C-terminal EF hands of the molecule are known to bind to various mechanosensitive proteins, such as palladin[22], titin[23]. We have previously shown that the dimer under unzip-force stretching geometry ruptures at forces of - 5 pN during linear force-increase scans with 1 pN s⁻¹[41] (also indicated by the red arrow in Fig. 3 with 5 pN s⁻¹). Here we designed a single-molecule construct of α-actinin 1 rod homo-dimer, where the biotinylated AviTag was placed at the N-terminus of the first SR rod and the SpyTag was placed at the C-terminus of the second SR rod, leading to an unzip-force stretching geometry when the molecule was stretched via these two tags (Fig. 3a). We further directly quantified the force-dependent lifetime of the α-actinin 1 homo-dimer under unzip-force stretching geometry by implementing a force–jump-cycle procedure (Fig. 3c–e). Once the dimer ruptures at forces <7 pN under unzip-force stretching, the SR domains remain folded, leading to a single step jump (Fig. 3c), corresponding to the release of the looped long flexible linker (Fig. 3a, inset). In sharp contrast to the dimer under shear–force geometry, the dimer's mechanical stability is considerably weaker by orders of magnitude, as evidenced by lifetimes spanning a mere few seconds above 4 pN (Fig. 3e).

### Comparable high mechanical stability of α-actinin homo- and hetero- dimers among isoforms

Subsequently, we aimed to investigate the mechanical stability of other isoforms of α-actinin. Figure 4 illustrates the force-induced rupture of the non-muscle isoform α-actinin 4 dimer, as well as its force-dependent lifetime under shear–force stretching geometry. Figure S9 illustrates the force-induced rupture of the hetero-dimer formed between α-actinin 1 and 4, along with its force-dependent

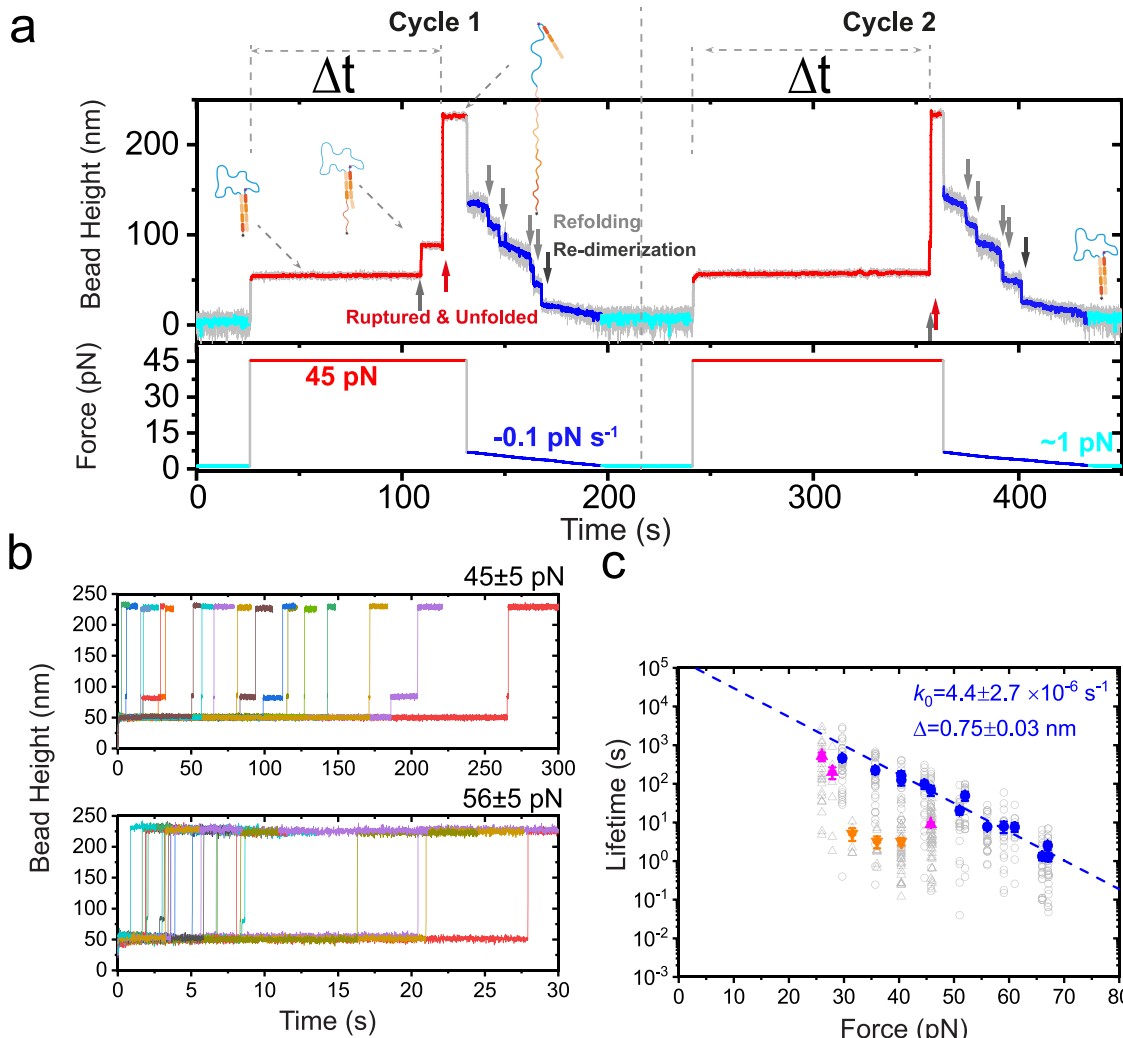

**Fig. 2 | Force-dependent lifetime of human α-actinin 1 homo-dimer under shear-stretching force geometry. a** The illustration and representative cycles of force–jump experiments for quantifying the force-dependent lifetime of the dimer. In each force–jump cycle, the force was first held at -1 pN (cyan) to ensure formation of the α-actinin 1 dimer, and then was jumped to a target force (e.g., 45 pN, red) and held at the force until rupture event was observed indicated by a large step jump (red arrow). After the rupture of the dimer (as well as the unfolding of the SR domains), the force was then fast decreased to -7 pN and slowly (−0.1 pN s⁻¹) decreased to 1 pN to observe the re-folding events of the four SR domains (gray arrows) and re-dimerization of the dimer (dark gray arrow), ensuring the molecule was in its dimerized state prior next scan cycle. In each scan, the dwell time of the dimer at the target force, Δ*t* was recorded. **b** The colored lines are representative time–bead height curves of the dimer at two target force examples (45 pN and 56

pN, respectively). The rupture event was indicated by the sudden large stepwise bead height jump at each scan. The theoretically predicated stepsizes of one SR unfolding and rupturing concurrent with another three SR unfolding are 30 nm and 138 nm at 45 pN, or 31 nm and 142 nm at 56 pN, respectively. **c** The force-dependent lifetime of the human α-actinin 1 homo-dimer under shear-stretching force geometry. The dwell times of the dimer at each force were plotted in gray hollow symbols, and the characteristic lifetime of each force was plotted in solid symbols. The blue line is Bell model[70] ($k_u(f) = k_0 \times \exp(\frac{\Delta f}{k_B T})$) fitting curve of the blue data sets, and gives $k_0$ and $\Delta$ indicated on the panel. Here we note that, in two sets of experimental data (magenta and orange colored symbols), much weaker mechanical stability was observed, and the data was excluded for Bell model fitting. Source data are provided as a Source Data file.

lifetime under shear–force stretching geometry. Both the homo-dimers and hetero-dimers of α-actinin 1 and 4 exhibit remarkable mechanical stability, demonstrating >100 s lifetime within 40 pN forces. Interestingly, it appears that the α-actinin 1 homo-dimer exhibits slightly greater mechanical strength than either the α-actinin 4 homo-dimer or the hetero-dimer counterpart.

## Mechanical states of α-actinin 1 homo-dimer with a SR1:SR4 pair ruptured

We aimed to conduct a more comprehensive investigation into the high mechanical stability exhibited by α-actinins. This involved a detailed analysis of the tandem spectrin repeats (SRs) to assess the mechanical resilience of specific domain pairs. Initially, we assessed the mechanical stability of the SR2–SR3–SR4:SR1–SR2–SR3 dimer. This

configuration mimics a scenario where one of the SR1:SR4 domain pairs within the α-actinin 1 homo-dimer is disrupted (refer to Fig. 5a, left panel). The corresponding force–bead height curves during a force-increase scan, conducted at a loading rate of 5 pN s⁻¹, are presented in the right panels of Fig. 5a–d. Additionally, the force–stepsize graph and rupture force distributions are depicted in Fig. 5e–f. Notably, the SR2–SR3–SR4:SR1–SR2–SR3 dimer reveals a range of mechanical states. The most robust mechanical state, characterized by a peak rupture force of 33 ± 9 pN (Fig. 5a, and red data in Fig. 5e, f), aligns with the weaker mechanical subgroup observed during the rupture of the full rod dimer (Fig. 1).

Unexpectedly, there are at least three additional, weaker mechanical groups observed within this structure. A plausible explanation is that in the SR2–SR3–SR4:SR1–SR2–SR3 dimer

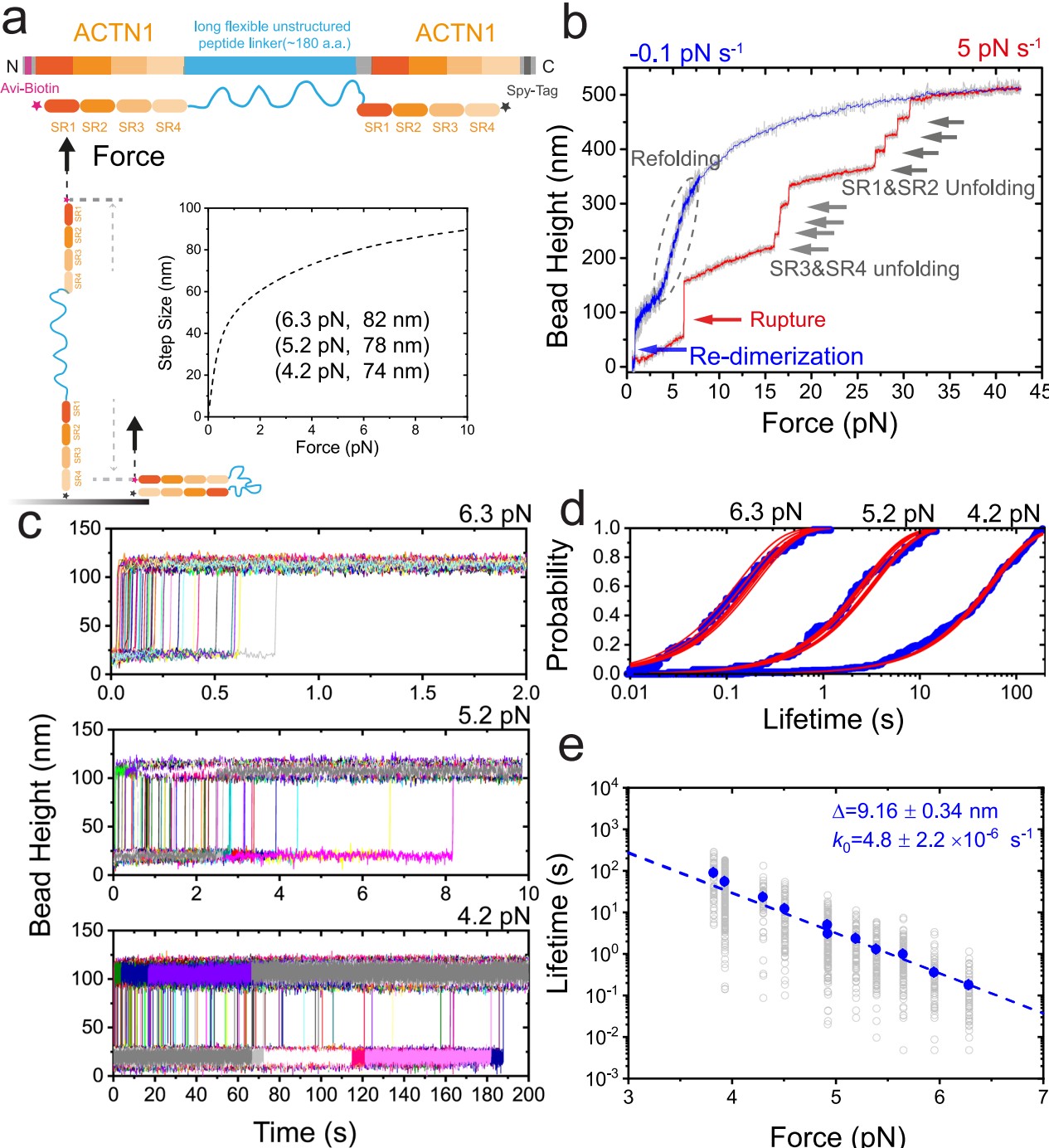

**Fig. 3 | Weak mechanical stability of the human α-actinin 1 homo-dimer under unzip-stretching force geometry. a** The illustration of the single-molecule construct and experimental design for quantifying the mechanical stability of the dimer under unzip-stretching force geometry. The biotinylated AviTag was placed at the N-terminus of the first rod, while the SpyTag was placed at the C-terminus of the second rod, resulting in unzip-force geometry when force was applied via these two tags. The inset is the theoretically predicated force-dependent rupture step-sizes of the dimer assuming the releasing of the looped linker under unzip-stretching geometry, while all SR domains remain folded. **b** The representative force–bead height curve of α-actinin 1 rod homo-dimer under unzip-stretching force geometry during force-increase scan (5 pN s⁻¹, red line) and force-decrease scan (−5 pN s⁻¹ to ~ 7 pN, and then −0.1 pN s⁻¹ to -1 pN, blue line). The rupture event is noted by a red arrow, the unfolding events of the eight domains are noted by

eight gray arrows, the re-dimerization event is noted by a blue arrow. The refolding events region is indicated by a dashed elliptic. **c** The colored lines are representative time–bead height curves of the dimer at three target force examples (bottom to top: 4.2 pN, 5.2 pN, and 6.3 pN, respectively). **d** The time evolution of the rupture probability under three example forces, from which the characteristic lifetime of the corresponding force was obtained by exponential fitting ($p(t) = 1 - \exp(-\frac{t}{\tau})$). **e** The force-dependent lifetime of the human α-actinin 1 homo-dimer under unzip-stretching force geometry. The dwell times of the dimer at each force were plotted in gray hollow symbols, and the characteristic lifetime of each force was plotted in solid symbols. The blue line is Bell model fitting curve of the blue data sets, and gives $k_0$ and Δ indicated on the panel. Source data are provided as a Source Data file.

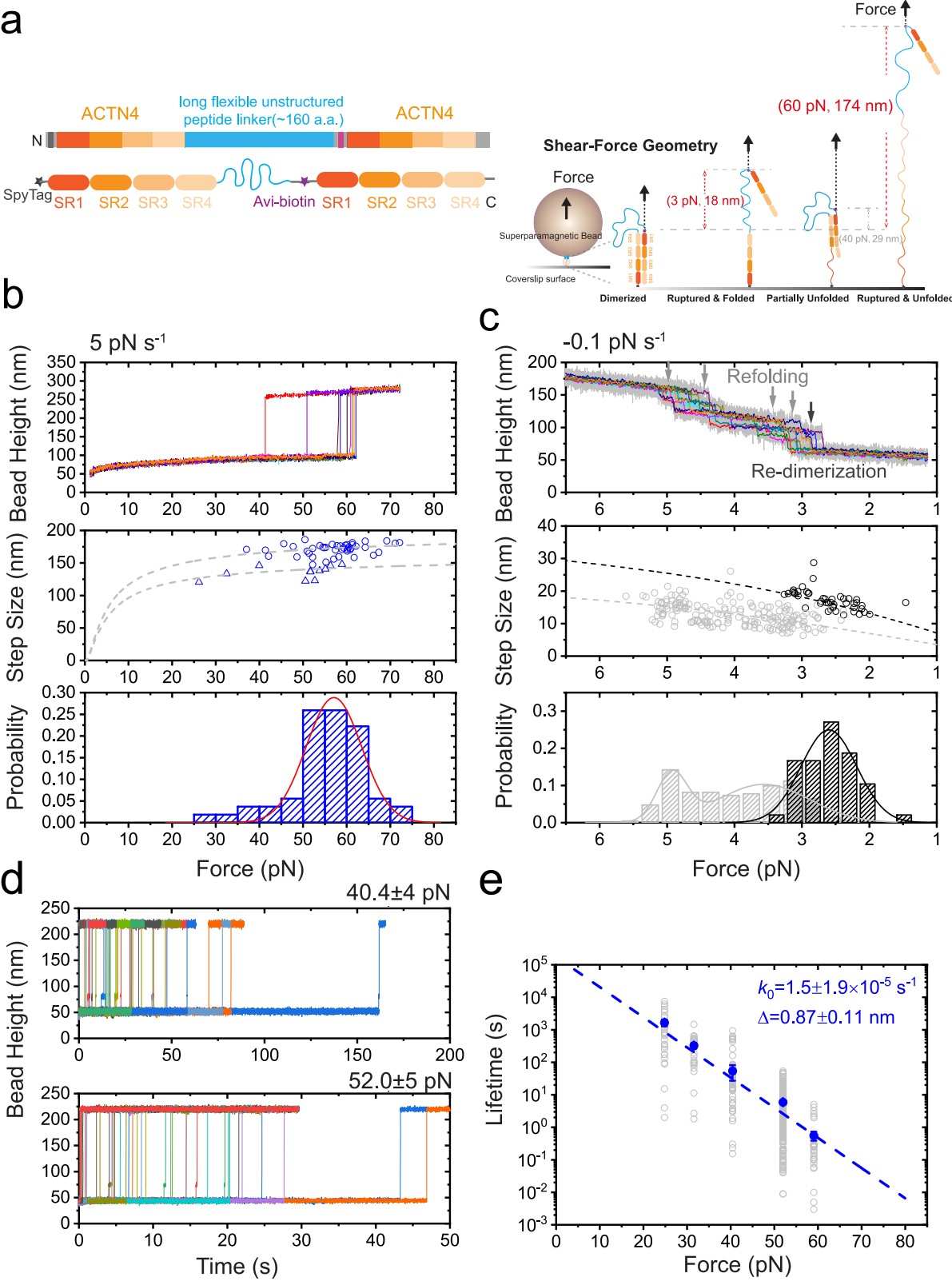

configuration, multiple subgroups of domain-pair states are associated with distinct mechanical stabilities. These subgroups can be categorized as follows: two of the SRs in the SR2–SR3–SR4 dimerize with their counterparts in SR1–SR2–SR3, while the remaining pair remains either undimerized or forms a weaker, non-native dimerized state (Fig. 5b, c, left panel). As the SR2–SR3–SR4 monomer unfolds within a force range of 15–25 pN (Fig. S11), this weakly

dimerized state can result in the rupture of the dimer either before or after the domain unfolding (Fig. 5b, c, right panel). Alternatively, one of the SRs in the SR2–SR3–SR4 dimerizes with its partner, leaving the other two pairs dimerized in a weaker, non-native state (Fig. 5d, left panel). Similar to the previous scenario, this weakly dimerized state can lead to the rupture of the dimer prior to or following the unfolding of the domain (Fig. 5d, right panel).

**Fig. 4 | High mechanical stability of the human *α*-actinin 4 homo-dimer under shear–force geometry. a** The illustration of the single-molecule construct and experimental design for quantifying the mechanical stability of the dimer. **b, c** Top panels: The representative force–bead height curves of *α*-actinin 4 rod homo-dimer under shear–force geometry during force-increase ($5\,\text{pN s}^{-1}$, left) and force-decrease ($-0.1\,\text{pN s}^{-1}$, right) scans, respectively. The gray arrows indicate the SR domains refolding, and the dark gray arrow indicates the re-dimerization event. The light gray and colored lines indicate the raw data and its 10-point FFT smoothing, respectively. Middle panels: The force–step size graph of the dimer during force-increase and force-decrease scans. Left middle panel: the dark gray and gray dashed lines are theoretically predicated force-dependent stepsizes assuming the concurrent rupturing and unfolding of all the four SR domains (dark gray) or with three SR domains (gray), respectively. $N = 54$. Right middle panel: the dark gray and gray dashed lines are theoretically predicated force-dependent stepsizes of the dimer formed by fully folded SR rods (dark gray) and individual SR domains (gray), respectively. re-dimerization events: $N = 48$; re-folding events: $N = 186$. Bottom panels: the normalized rupture forces distribution obtained during force-increase scans with $5\,\text{pN s}^{-1}$ (left), and the re-dimerization forces distribution (dark gray) and refolding force distribution (gray) obtained during force-decrease scans with $-0.1\,\text{pN s}^{-1}$ (right). The lines are gaussian fitting curves of the distributions, with the peak values (from left to right) as $57.1 \pm 0.7$ pN, $4.92 \pm 0.04$ pN, $3.56 \pm 0.11$ pN and $2.60 \pm 0.05$ pN, respectively. **d** The colored lines are representative time–bead height curves of the dimer at two example forces (40 pN and 52 pN, respectively) obtained by force–jump cycle procedures. **e** The force-dependent lifetime of the dimer under shear–force geometry. Gray hollow symbols and solid symbols indicate the dwell times and the characteristic lifetime of the dimer at each force, respectively. The blue line is Bell model fitting curve of the blue data sets, and gives $k_0$ and $\Delta$ indicated on the panel. Source data are provided as a Source Data file.

An intriguing observation is the clustered appearance of these subgroups within the time trace during repeating force scan cycles. This clustering phenomenon persists for several to tens of cycles before transitioning from one group to another. Such clustered behaviors were leveraged to measure the force-dependent lifetimes of these subgroups. The clustered mechanical groups might involve multiple intermediate conformational states of the dimer associated with certain energy barriers, which will be elaborated in the discussion section.

Figure 5g illustrates the force–jump-cycle procedure for quantifying the force-dependent lifetimes of the construct. Figure 5h shows the resulting force-dependent lifetime of the SR2–SR3–SR4:SR1–SR2–SR3 dimer. In line with the distinct rupture forces, these sub-groups display varying force-dependent lifetimes, differing by orders of magnitude. Inter-conversions between these sub-groups were also noted at a constant force of approximately 2.5 pN over a span of 60,000 s. This was indicated by three clearly distinguishable clusters representing different lifetimes of the dimer (Fig. S10). Given that all SR domains are in a folded state at this force, the rupture of the dimer leads to a single-step bead height jump without the SRs unfolding.

**Mechanical states of *α*-actinin 1 homo-dimer with both SR1:SR4 pairs ruptured**

Subsequently, we investigated the mechanical stability of the dimer formed by SR2–SR3:SR2–SR3, which emulates a scenario in which both SR1:SR4 domain pairs are disrupted (Fig. 6a). The force–bead height curve of the SR2–SR3:SR2–SR3 construct during force-increase scans, conducted at a loading rate of $1\,\text{pN s}^{-1}$, is depicted in top panel of Fig. 6b. Intriguingly, the dimeric pair dissociated at forces of approximately 2–4 pN, followed by the unfolding of the stretched SR3 and SR2 domains at around 15 pN and 25 pN, respectively. During force decreasing (with a loading rate of $-0.1\,\text{pN s}^{-1}$), the unfolded SR3 and SR2 refolds at ~4 pN, and enables the re-dimerization of the SR2–SR3:SR2–SR3 pair at forces ~3 pN (bottom panel, Fig. 6b). These findings indicate that the SR2–SR3:SR2–SR3 dimer can form a mechanically weak interaction that readily unfolds at forces exceeding 2 pN within seconds (Fig. 6c–f). This mechanical characteristic is akin to the subgroup displaying low mechanical stability observed in the SR2–SR3–SR4:SR1–SR2–SR3 dimer (orange data in Fig. 5).

In addition, we explored the mechanical stability of a potential dimer formed by a single SR1:SR4 pair under shear–force stretching geometry. Notably, only the SR1 unfolding event was detected during each force-increase scan (Fig. S12), indicating that the potential dimer formed by a single SR1:SR4 pair was mechanically unstable within the measured force range ($\geq 1$ pN).

**The mechanical regulation of the ancestral *Entamoeba histolyticaα*-actinin 2 homo-dimer**

The phylogenetic analysis of the *α*-actinin rod domain suggests that modern vertebrate *α*-actinins have evolved from ancestral forms containing one or two SR domains through intragenic duplication[59,60]. In this context, the fungal isoforms of *α*-actinin and the protozoan *α*-actinin represent a significant evolutionary stage, featuring proteins with two SR repeats. This prompts an inquiry into how the mechanical stability of *α*-actinins evolved over history. Consequently, we investigated the mechanical stability of the model ancestral *α*-actinin, namely *Entamoeba histolyticaα*-actinin 2 (a protozoan variant) (Fig. 7a)[61].

In Figure 7b, we present the force–bead height curves that depict the representative behavior of the *Entamoeba histolyticaα*-actinin dimer during force-increase scans (using a loading rate of $5\,\text{pN s}^{-1}$). Notably, the *α*-actinin dimer exhibits multiple mechanical states (three states were observed, indicated by blue, magenta and orange arrows) that undergo rupture at distinct forces during force-increase scans. The mechanically strongest state, characterized by a peak rupture force of $33 \pm 9$ pN ($5\,\text{pN s}^{-1}$), demonstrates a lifetime of >100 s at forces <15 pN (Fig. 7e, blue). The second strong state, with a peak rupture force of $20 \pm 9$ pN ($5\,\text{pN s}^{-1}$), maintains a lifetime of >10 s at forces <10 pN (Fig. 7e, magenta). The third state, which ruptures at approximately 5 pN, exhibits a lifetime of >1 s at forces <10 pN (Fig. 7e, orange). Notably, inter-conversions among these three subgroups were also observed at a constant force of ~2.4 pN over a span of 60000 seconds, as indicated by distinct clusters of lifetimes in the dimer (Fig. 7d). Altogether, the ancestral *Entamoeba histolyticaα*-actinin homo-dimer formed with two SR domain pairs also shows weak-to-strong mechanical stabilities in a clustered manner.

## Discussion

In this work, we systematically probed the mechanical stabilities of human *α*-actinin dimers within a physiologically relevant force range at single-molecule level. We showed that the homo-dimers formed by *α*-actinin 1 or 4, and the hetero-dimer of *α*-actinin 1 and 4 all have a high mechanical stability under shear-stretching force geometry. The rupture of the dimers often occurs at forces $\geq 60$ pN with a physiologically relevant force-loading rate of $5\,\text{pN s}^{-1}$. The *α*-actinin dimers have a lifetime of $\geq 100$ s within a physiologically relevant force range of $\leq 40$ pN. Such a high mechanical stability of the *α*-actinin dimers under shear-stretching force geometry ensures their role as the major and strong actin filament crosslinker, compared to the another actin filament crosslinker protein filamin, which is recently reported to be ruptured at forces of ~14 pN[52].

In sharp contrast, the *α*-actinin dimer has an over orders weaker mechanical stability under unzip-stretching force geometry. The rupture of the dimer often occurs at forces ~6 pN (with a force-loading rate of $5\,\text{pN s}^{-1}$), and is associated with a lifetime $\leq 10$ s with forces $\geq 5$ pN. Such a distinct mechanical stability of the *α*-actinin dimer under shear-stretching or unzip-stretching force geometry might contribute to the dynamics and versatility of *α*-actinin dimers.

We also showed that the high mechanical stability of *α*-actinin dimer under shear-stretching force geometry is built-up by the four SR domain pairs from each monomer, i.e., SR1–SR2–SR3–SR4:SR4–SR3–SR2–SR1.

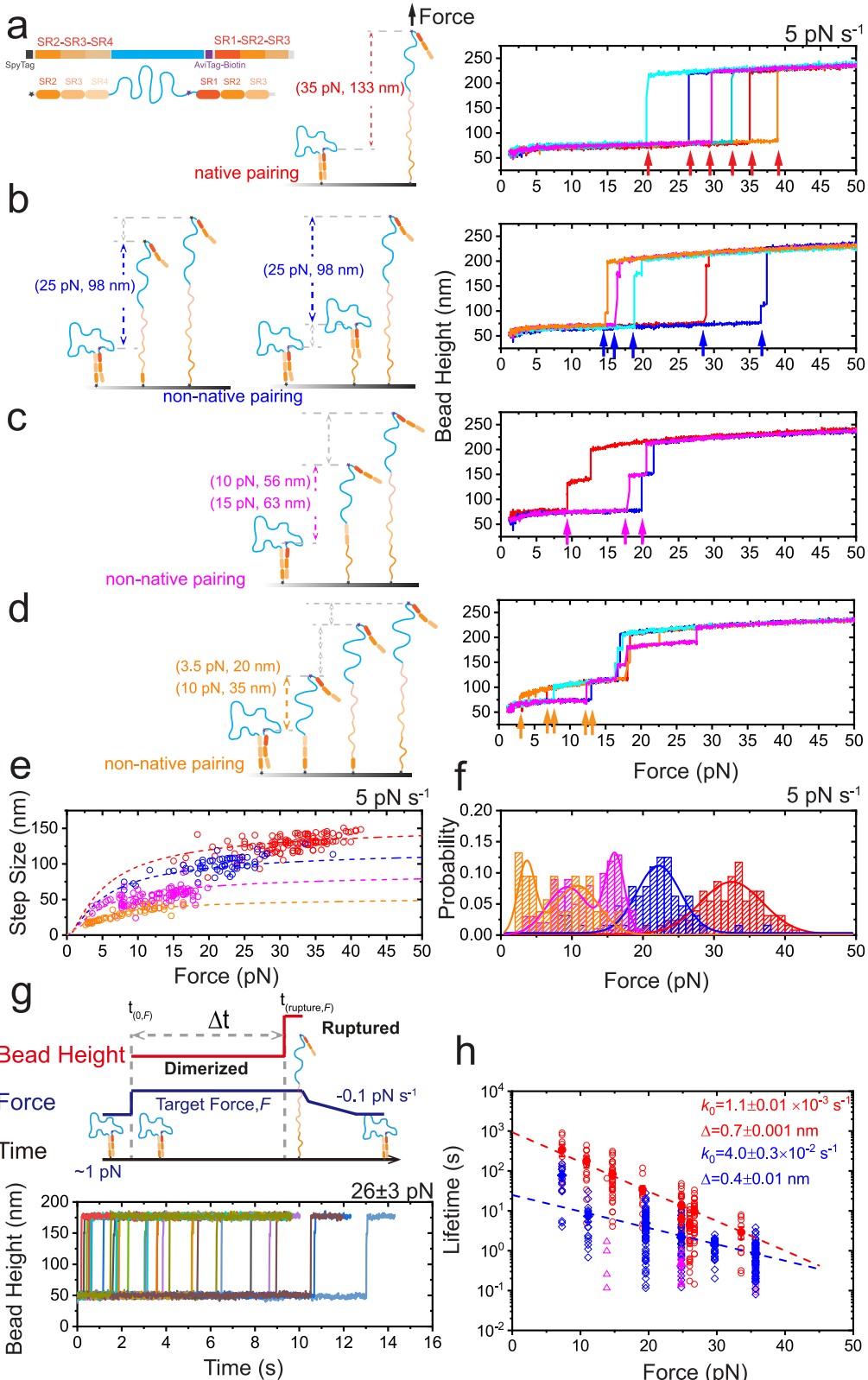

When one of the SR1:SR4 pair is un-dimerized, the force-dependent lifetime of the dimer formed by the remaining domain pairs decreases about 1000-folds compared to the full rod dimer. It ruptures at forces of ~35 pN, which is about 30 pN less than the full dimer. Nevertheless, such a weakened partial dimer with one SR1:SR4 pair un-dimerized still has a sufficient mechanical stability within a physiologically relevant force range. It has a lifetime of ≥100 s with forces ≤15 pN. This result suggests that the α-actinin dimer could act as strong mechanical crosslinker on actin filaments while allowing its SR1:SR4 sub-domain pair being un-dimerized. Since an α-actinin monomer contains a single cryptic vinculin binding site buried in the folded SR4 domain[41,62]. The ability of being partially un-dimerized and unfolded of the SR1:SR4 on the dimer allows the α-actinin dimer signaling with vinculin while still crosslinking the actin filaments. Furthermore, even when both the

**Fig. 5 | Mechanical stability of partial α-actinin 1 homo-dimer of SR2–SR3–SR4:SR1–SR2–SR3 under shear–force geometry. a–d** Representative force--bead height curves of partial α-actinin 1 rod homo-dimer of SR2–SR3–SR4:SR1–SR2–SR3 under shear–force geometry during force-increase scans (5 pN s⁻¹). Four distinct rupture behaviors were plotted in right panels (**a–d**), respectively. The distinct rupture events are indicated by different colored (red, blue, magenta, orange) arrows. Left panels: illustrations of possible conformational changes of partial dimers associated with different pairing states. The colored dashed lines in the left panels indicate the height jump when the rupture event occurs. **e** The force–step size graph of the rupture events of SR2–SR3–SR4:SR1–SR2–SR3 dimer during force-increase scans. the four colored dashed lines are the theoretically predicated force-dependent stepsizes assuming the concurrent rupturing and unfolding of all the three SR domains (red), with two SR domains (blue), with one SR domains (magenta), or without SR domain (orange), respectively. $N = 128$ (red), 64 (blue), 101 (magenta) and 51 (orange). **f** The normalized rupture force distribution of the SR2–SR3–SR4:SR1–SR2–SR3 dimer during force-increase scans. The lines are gaussian fitting curves of the distributions, with the peak values as $32.5 \pm 0.3$ pN (red), $22.2 \pm 0.2$ pN (blue), $16.1 \pm 0.2$ pN and $9.4 \pm 0.3$ pN (magenta), $10.8 \pm 0.5$ pN and $3.6 \pm 0.2$ pN (orange) respectively. **g** Top panel: The illustration of the experimental design for quantifying the force-dependent lifetime of the partial dimer. Bottom panel shows the representative time–bead height curves of the partial dimer at an example target force of 26 pN. **h** The force-dependent lifetime of the α-actinin 1 partial dimer of SR2–SR3–SR4:SR1–SR2–SR3 under shear–force geometry. Colored hollow symbols and solid symbols are the individual dwell times of the dimer and the characteristic lifetime of the dimer at each force, respectively. Three clustered force-dependent lifetime relations were obtained indicated by three colored symbols, respectively. Two sets of the lifetime were fitted with Bell Model (red and blue colored lines, respectively) to obtain the corresponding $k_0$ and $\Delta$. The magenta data set was not fitted due to limited data obtained. Source data are provided as a Source Data file.

SR1:SR4 pairs are un-dimerized, the dimer formed by the remaining domain pairs (the two SR2:SR3 pairs) could still withstand forces of a few pN, suggesting a potentially more dynamic and versatile role of cross-linking and signaling of the α-actinin dimer under force. How the dynamic force-dependent conformational states of the SR domains modulate their interactions with a number of cytoskeletal and regulatory proteins[8], such as zyxin, ICAM-1/2, warrant future studies.

Since the modern vertebrates' α-actinins are evolved from the ancestral α-actinins that contains one or two SR domains by intra-genic duplication[59,60], we also probed the mechanical stability of an ancestral α-actinin, the *Entamoeba histolytica* (protozoan)α-actinin 2, and showed that the ancestral two subdomain pair α-actinin homo-dimer also have multiple mechanical lifetimes, including a high mechanical stability with a peak rupture force of $33 \pm 9$ pN (5 pN s⁻¹) and is associated with a lifetime of >100 s at forces <15 pN, and weaker mechanical stabilities associated with orders shorter lifetime at same force range. The results suggest an evolutional conservation of the weak-to-strong mechanical strength-building mechanism of the α-actinin dimers.

The modern α-actinin dimer leverages its four SR domain pairs to enhance its mechanical strength, conferring distinct mechanical properties and functional versatility to α-actinin dimers. We posit that the multivalent interaction kinetics of the dimer might be a primary molecular mechanism underlying the dimer's strength enhancement. This model involves a dimer with two proteins, each comprising two to four domains that pair to corresponding domains on the other protein, resulting in a multivalent nature of the dimer (Fig. 8).

In the model considering multivalent interaction kinetics of the dimer, the dissociation transition path of the dimer is anticipated to initiate from the force-bearing ends under mechanical force (Fig. 8). Since the dissociated domains remain in proximity due to the remaining of dimerized domain pairs, rapid re-association of the dissociated domain becomes possible, leading to stabilization through multivalency. Additionally, in the dissociated state, one domain experiences force, potentially inducing unfolding and refolding of the force-bearing domain in a force-dependent manner. The unfolding of the domain hinders re-association. Consequently, the force-dependent lifetime of this protein–protein complex is governed by the force-dependent dissociation and re-association rates of the boundary domain pairs, the number of interacting domain pairs in the complex, as well as the force-dependent unfolding and refolding of the dissociated domains at the boundary (see Fig. 8). The inherent multivalency of the multiple domain dimer leads to a substantial increase in lifetimes. Here we note that the lifetime enhancement effect due to multivalency applies note only for multiple domain dimer, but also in general inter-molecular interactions with multiple interacting sites.

While the above model considering multivalent interaction kinetics explains the enhancement of mechanical stability as domain pair number increases, it alone could not explain the clustered effects of the mechanical stabilities observed in the α-actinin dimers. The rigidity of the rod and 90° twisted conformation of the α-actinin dimer[7,15] may hint on additional molecular mechanisms involving the rigid domain pair re-orientation. The SR rod monomer contains four SR domains linked by a short helical linker. The linker between two SR domains is a short rigid helix extended from the C-terminal helix of the former SR or the N-terminal helix of the later SR[7,15]. When the SR rods form dimers, the SR domains in the rod not only need to be paired, but also need to be re-oriented to suit into the native 90° twisted conformation[7,15]. Due to the rigidity of the helical linker and the domains, the domain-pairs may be trapped in certain intermediate conformational states where some of the domains pair into the native states, while some of the domains pair and re-orientate in non-native states, leading to the clustered distinct mechanical stability group. On the other hand, the domain-pair re-orientation model also implies that natively re-oriented domain pairs may facilitate their neighboring domain pairs' re-orientation, leading to cooperativity of the domain-pairs re-orientation.

Consequently, the four-SR dimer shows primarily the strong mechanical group, while the three-SR dimer shows at least three additional distinct mechanical groups due to potential intermediate pairing states of domain pairs. Consistently, the two-SR dimer of human α-actinin dimer only shows one weak mechanical group due to weakened re-orientational cooperativity and less domain interaction kinetics. In sharp contrast, the two-SR dimer of *Entamoeba histolytica* actinin 2 shows at least three distinct mechanical group. It is possible due to that while the *Entamoeba histolytica* actinin 2 dimer only contains two SR dimer, it also re-orientated about 90° from one end to another[61], leading to re-orientational cooperativity during dimerization. Further investigations combining molecular dynamic simulations, FRET-based protein dynamics, etc. on the dimer's domain pairing and re-orientational dynamics are warranted.

In addition, the design of the single-molecule experiments and theoretical framework could also be implemented for systemically studying the mechanical regulation of other multiple domain homo- or hetero- dimers, such as cadherin dimers, which are formed with five sub-domain pairs[4,14,63–65]. The experiments could be further expanded to investigate the mechanically regulated protein–protein interactions by introducing corresponding singling proteins in the solution and quantitatively probe the effects on the mechanical stability and conformation of the multi-domain dimers under force.

## Methods
### Protein constructs list
Eight plasmids were prepared for expression of the protein constructs for single-molecule stretching experiments, and a SpyCatcher-containing plasmid was prepared for expression of the SpyCatcher protein. The detailed sequences of the corresponding protein constructs are listed in Supplementary Note S1:

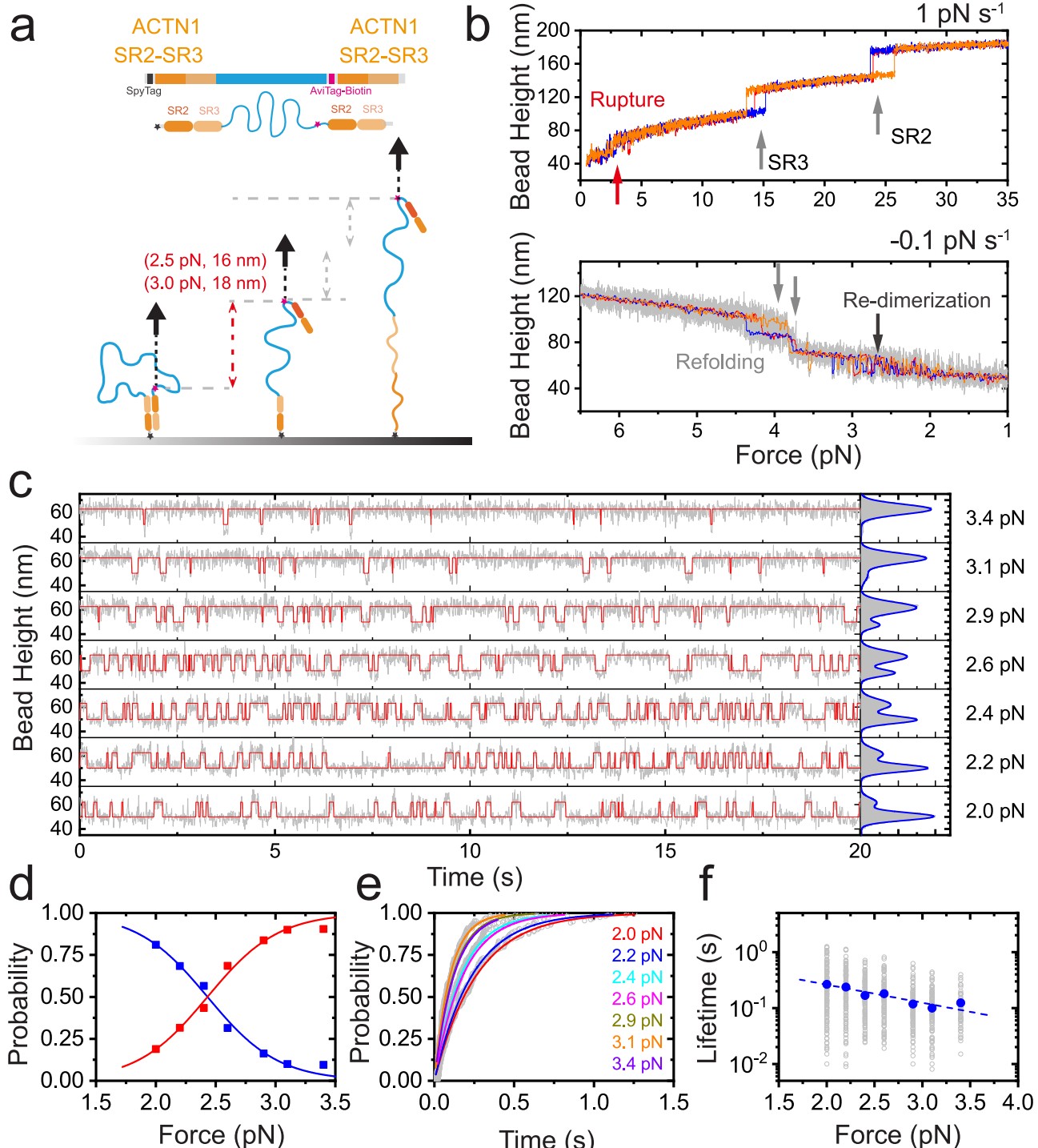

**Fig. 6 | Mechanical stability of a partial α-actinin 1 homo-dimer of SR2–SR3:SR2–SR3 under shear-stretching force geometry. a** The illustration of the single-molecule construct and experimental design for quantifying the mechanical stability of the partial dimer. **b** Representative force–bead height curves of partial α-actinin 1 rod homo-dimer of SR2–SR3:SR2–SR3 under shear-stretching force geometry during force-increase scans (1 pN s⁻¹, top panel) and force-decrease scans (−0.1 pN s⁻¹, bottom panel). The rupture and re-dimerization events are noted by the red arrow and dark gray arrows, respectively. The unfolding and refolding events of the domains are noted by gray upward arrows and down-ward arrows, respectively. **c** Representative time–bead height curves of the partial dimer at forces in a range of 2.0 pN to 3.4 pN in 20 s time window. The red lines are the step-wise fitting of the raw data in gray by hidden Markov model. The right

panel shows the normalized bead height distributions at each force obtained by 200 seconds data recording. The blue lines are gaussian fitting of the distributions. **d** The force-dependent rupture probability (red) and re-dimerization probability (blue). The symbols are experimental data obtained. The blue line is obtained by fitting with $P_{\mathrm{re-dimerization}}(F) = 1/\left(\exp\frac{(F-F_c)\Delta x}{k_B T} + 1\right)$; the red line is obtained by fitting with $P_{\mathrm{rupture}}(F) = 1 - P_{\mathrm{re-dimerization}}(F)$. **e** The time evolution of the rupture probability at each force, from which the characteristic lifetime of the corresponding force was obtained by exponential fitting with $P(t) = 1 - \exp(-\frac{t}{\tau})$. **f** The resulting force-dependent lifetime of the partial dimer. The dwell times of the dimer at each force were plotted in gray hollow symbols, and the characteristic lifetime of each force was plotted in solid symbols. The blue dashed line is Bell model fitting curve of the blue data set. Source data are provided as a Source Data file.

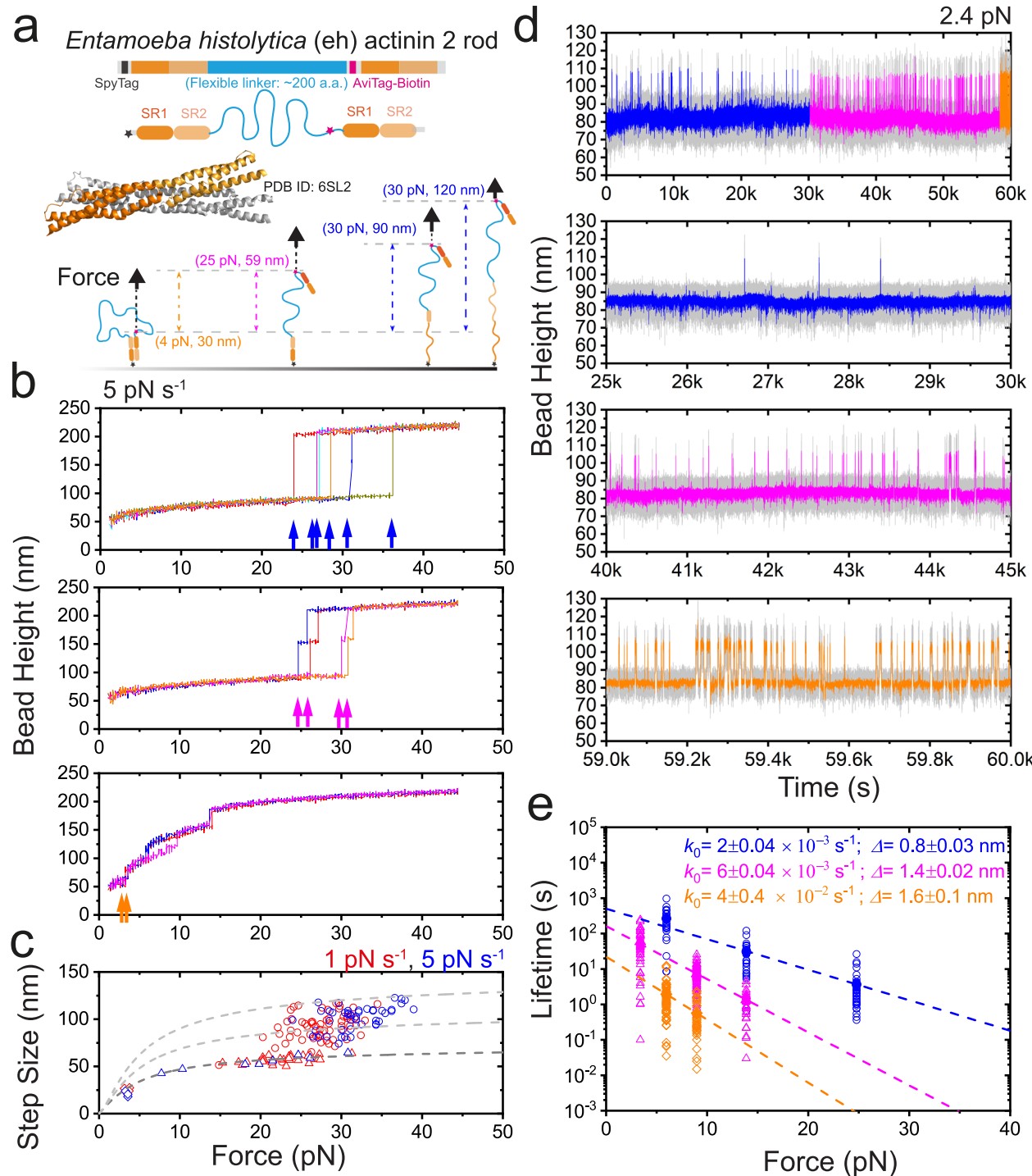

**Fig. 7 | Mechanical stability of an ancestral *Entamoeba histolytica*α-actinin homo-dimer under shear-stretching force geometry. a** The illustration of the single-molecule construct and experimental design for quantifying the mechanical stability of the *Entamoeba histolytica*α-actinin dimer. Three possible scenarios of rupture-induced bead height changes are indicated by colored dashed arrows: rupture without domain unfolding (orange), rupture with one domain unfolding (magenta) and with two domain unfolding (blue). **b** Representative force–bead height curves of *Entamoeba histolytica*α-actinin dimer under shear-stretching force geometry during force-increase scans (5 pN s⁻¹). Three distinct rupture behaviors were observed and plotted in top-to-bottom panels, respectively. The rupture events at each panel are noted by colored arrows. **c** The resulting force–step size graph of the rupture events during force-increase scans at two force-loading rates of 1 pN s⁻¹ (red) and 5 pN s⁻¹ (blue), respectively. The different mechanical groups

are plotted with different symbols (circle, triangle, and diamond). *N* = 99 (red) and 56 (blue), respectively. **d** Top panel: inter-conversions of the three subgroups at constant forces of ~2.4 pN over 60,000 s. Second to bottom panels: the zoom-in of the time−bead height curves of the three subgroups, indicated by different colors. **e** The force-dependent lifetime of the *Entamoeba histolytica*α-actinin dimer under shear-stretching force geometry. The dwell times of the dimer at each force were plotted in colored hollow symbols, and the characteristic lifetime of each force was plotted in solid colored symbols. Three clustered force-dependent lifetime subgroups were obtained indicated by three colored symbols, respectively. Three sets of the lifetime were fitted with Bell Model (blue, magenta and orange colored lines, respectively) to obtain the corresponding $k_0$ and Δ. Source data are provided as a Source Data file.

## Multi-domain interaction kinetics under force

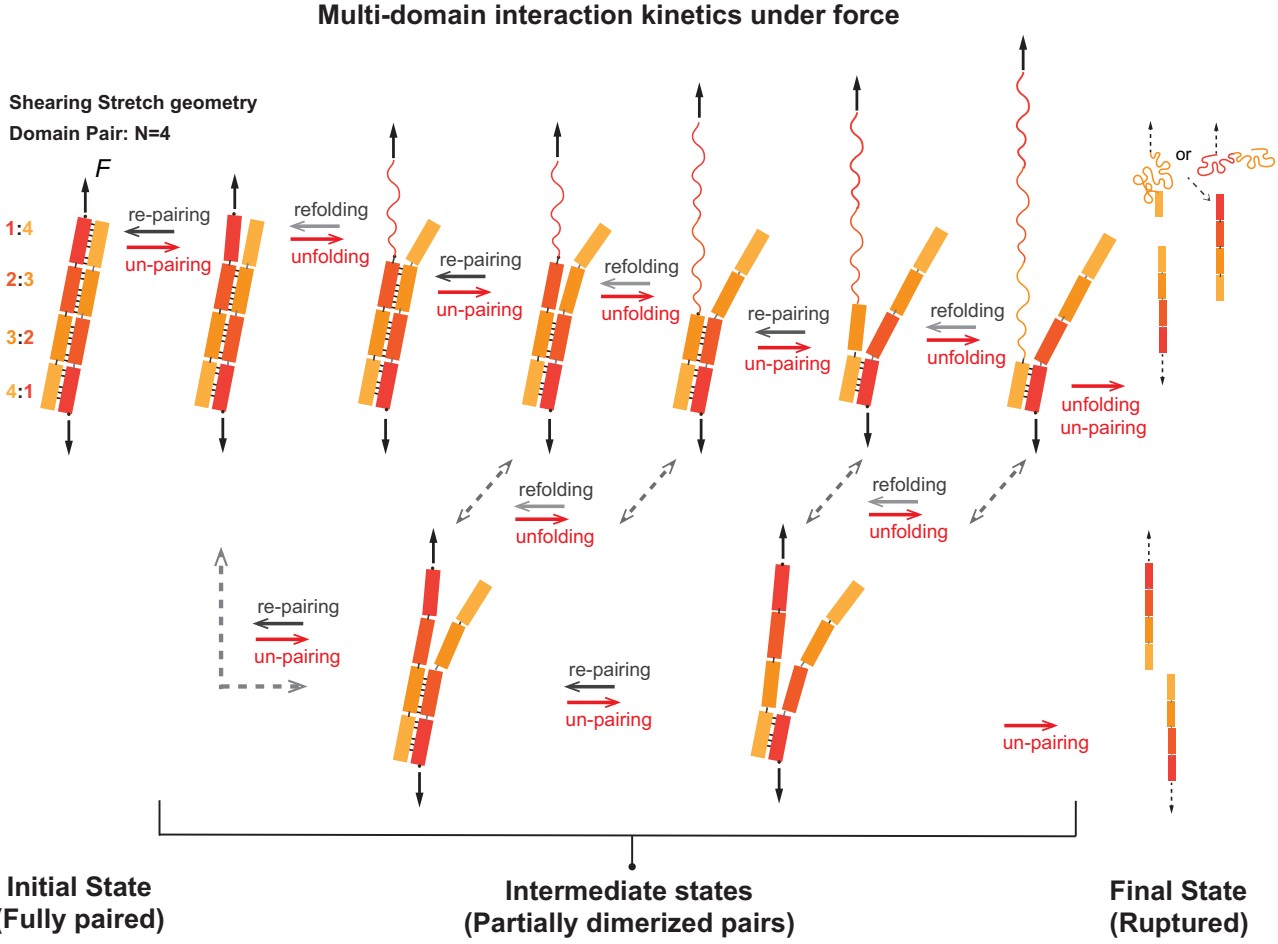

**Fig. 8 | Multi-domain interaction kinetic model of dimer strength-building.** Illustration of multivalent interaction kinetics of a dimer model with 4 domain pairs, revised based on ref. 4. In the model, the dissociation transition path of the dimer is anticipated to initiate from the force-bearing ends under mechanical force. Since the dissociated domains (un-pairing) remain in proximity due to the remaining of dimerized domain pairs, rapid re-association (re-pairing) of the dissociated domain becomes possible, leading to stabilization through multivalence.

Additionally, in the un-paired state, the stretched domain experiences force, potentially inducing force-dependent unfolding and refolding of the force-bearing domains. The domain unfolding hinders re-pairing of the domains. Consequently, the force-dependent lifetime of this protein–protein complex is governed by the force-dependent un-pairing and re-pairing rates of the boundary domain pairs, the number of interacting domain pairs in the complex, as well as the force-dependent unfolding and refolding of the unpaired domains at the boundary.

1. pET151–Spy003–(SR1–SR2–SR3–SR4)$_{ACTN1}$–(GS Linker)–AviTag-(SR1–SR2–SR3–SR4)$_{ACTN1}$
2. pET151–Spy003–(SR1–SR2–SR3–SR4)$_{ACTN4}$–(GS Linker)–AviTag-(SR1–SR2–SR3–SR4)$_{ACTN4}$
3. pET151–Spy003–(SR1–SR2–SR3–SR4)$_{ACTN1}$–(GS Linker)–AviTag-(SR1–SR2–SR3–SR4)$_{ACTN4}$
4. pET151–AviTag–(SR1–SR2–SR3–SR4)$_{ACTN1}$–(Long Linker)–(SR1–SR2–SR3–SR4)$_{ACTN1}$–Spy
5. pET151–Spy003–(SR2–SR3–SR4)$_{ACTN1}$–(GS Linker)–AviTag-(SR1–SR2–SR3)$_{ACTN1}$
6. pET151–Spy003–(SR2–SR3)$_{ACTN1}$–(GS Linker)– AviTag–(SR2–SR3)$_{ACTN1}$
7. pET151–Spy003–(SR1)$_{ACTN1}$–(GS Linker)–AviTag–(SR4)$_{ACTN1}$
8. pET151–Spy003–(SR1–SR2)$_{Entamoebahistolytica}$–(GS Linker)–AviTag–(SR1–SR2)$_{Entamoebahistolytica}$
9. pET151–SpyCatcher–Cys

### Plasmids cloning and protein expression
Briefly, the DNA fragments encoding the domains were amplified by PCR using the template sequences[41]. The SpyCatcher[55,56] DNA fragment was synthesized by IDTgblock. The long flexible linker and other DNA fragments were synthesized by GeneArt/IDTgblock. These DNA fragments were then assembled into a pET151 plasmid template[41] using

HiFi DNA Assembly (NEB) with DH5α competent cells. The resulting plasmids were confirmed by sequencing.

Each α-actinin plasmid was co-transformed with a BirA-containing plasmid and expressed in *Escherichia coli* BL21 (DE3), cultured in LB-broth media with D-Biotin (Sigma Aldrich), and then affinity-purified through a 6His-tag. The SpyCatcher plasmid was transformed, expressed in *Escherichia coli* BL21 (DE3), cultured in LB-broth media, and affinity-purified through a 6His-tag as well. For each α-actinin plasmid, after co-transformation and overnight culture on a LB-agar plate at 37 °C, a single colony was selected and cultured in 5 mL LB-broth media overnight at 37 °C with a shaking speed of 250 rpm. The cultured media were then transferred to new LB-broth media at a 1:20 ratio and continued to be cultured for approximately 3 h at 37 °C until the optical density (OD) reached around 0.5. Subsequently, IPTG and D-Biotin were added with final concentrations of 0.4 mM and 50 μM, respectively. The media were then cultured at 20 °C with a shaking speed of 250 rpm for approximately 16 h. The bacteria were collected by centrifuging the media at a speed of 4000 rpm, lysed by sonication, and then purified using a protein purification kit (Invitrogen/Sangon). Imidazole used during purification was removed by buffer exchange through dialysis or by using the AKTA system. The final stocking buffer comprises 50 mM Tris (pH 7.4), 500 mM NaCl, 5 mM DTT, and 10% glycerol. The final proteins

were aliquoted, snap-frozen with liquid nitrogen, and stored in a −80 °C freezer.

## Single-protein manipulation and analysis

A vertical magnetic tweezers setup was combined with a disturbance-free, rapid solution-exchange flow channel to conduct in vitro protein stretching experiments[53,54,66]. All in vitro protein stretching experiments were conducted in a solution containing 1× PBS, 1% BSA, 1 mM DTT, and 10 mM sodium L-ascorbate at a temperature of 22 ± 1 °C. The force calibration of the magnetic tweezers setup has an uncertainty of 10% due to the heterogeneity in the diameter of paramagnetic beads[53]. The temporal and spatial resolution of the magnetic tweezers setup were 200 Hz and 1 nm, respectively. The determination of bead height in the magnetic tweezers setup has an uncertainty of approximately 2–5 nm due to thermal fluctuations of the tethered bead and the camera's resolution[51]. The preparation of the sample channel and the methods for data analysis have been previously described[51,67,68], and are also briefly outlined below.

## Surface preparation

A coverslip (22 mm × 40 mm in width and length, with a thickness of 0.13–0.17 mm) was employed as the bottom coverslip of the channel. The coverslip underwent a cleaning process comprising sonication with deionized (DI) water (10 min), detergent solution (30 min), DI water (10 min), methanol (10 min), and DI water (10 min). Subsequently, it was rinsed with DI water, dried in an oven (100 °C), plasma activated (10 min), and immediately treated with 3-(Triethoxysilyl)propylamine (APTES, 1% in methanol) for 1 h. Following this treatment, the coverslip was sonicated, washed with DI water, and dried in an oven (100 °C).

To construct the sample channel, an APTES-coated bottom coverslip was paired with a cleaned top coverslip (20 mm × 20 mm in width and length, with a thickness of 0.13–0.17 mm). These two coverslips were separated by biocompatible double-sided tape with a thickness of 240 μm. The sample channel underwent two treatments: firstly, it was exposed to glutaraldehyde (1% in 1× PBS) for 2 h at 22 °C, and secondly, it was incubated with SpyCatcher-cys (0.05 mg mL$^{-1}$ in 1× PBS) for 6 h at the same temperature. Finally, the channel was incubated with a BSA solution (3% BSA in 1× PBS) for 24 h at 22 °C. The BSA-blocked SpyCatcher-coated channel was then stored at 4 °C and remained usable for up to 6 months. Here we note that the reference polystyrene beads were coated onto the surface after the glutaraldehyde treatment but prior to the SpyCatcher treatment step.

## Protein tether formation

To tether specific target protein molecules between the bottom coverslip surface and superparamagnetic beads within a sample channel, the target protein molecules (labeled with specific biotin and SpyTag003) were initially diluted to approximately 10$^{-4}$ mg mL$^{-1}$ in a standard buffered solution. They were then introduced into the SpyCatcher-coated sample channel, facilitating the specific SpyTag003-SpyCatcher interaction during an incubation period of 5–20 min at 22 °C. Subsequent to this interaction, unbound molecules were washed away using the standard buffered solution. The streptavidin or neutravidin coated super-paramagnetic beads (2.8 μm in diameter, Invitrogen M270) was diluted into ~0.1 mg mL$^{-1}$ in standard solution and incubated in the channel for 5–10 min (22 °C) to allow the specific interaction of the biotin on the surface-tethered protein with the streptavidin or neutravidin on the bead. The unbound beads were then gently washed way.

For the tethering of specific target protein molecules between the bottom coverslip surface and superparamagnetic beads with a DNA handle in between, after the incubation and washing steps of the target protein molecules, streptavidin or neutravidin at a concentration of 10$^{-2}$ mg mL$^{-1}$ in a standard buffered solution was introduced into the channel. This mixture was allowed to incubate

for approximately 10 min at 22 °C, facilitating binding between the streptavidin or neutravidin and the biotin on the surface-tethered protein. The unbound streptavidin or neutravidin were then washed away. Prepared superparamagnetic beads (2.8 m in diameter, Invitrogen M270) linked with biotinylated 572-bp double-stranded DNA were diluted to approximately 0.1 mg mL$^{-1}$ in the standard solution. These beads were then incubated within the channel for 5–10 min at 22 °C, promoting a specific interaction between the biotin on the DNA handle and the streptavidin or neutravidin on the target protein. Following the incubation, any unbound beads were gently washed away.

## Bead-height change in the magnetic-tweezer setup

In the magnetic tweezer experiments, a single target protein was tethered between a 2.8-μm-diameter superparamagnetic bead (with or without DNA handle) and the bottom coverslip surface. What we recorded was the height of the bead from the coverslip surface, aligned with the force direction. During a force change, the height change of the bead encompassed contributions from both extension change of the molecule and the bead re-orientation due to torque rebalance subsequent to the force change (Fig. S2). Hence, a force jump (a process typically completed within ≤0.25 s in our configuration) was typically accompanied with a stepwise bead height change, which magnitude depends on the level of bead rotation due to torque rebalance and the extension change of the molecule.

Conversely, when maintaining a constant force, torque equilibrium remains unaltered. Therefore, the bead height change at a constant force equals to the extension change of the molecule. During force-increase/decrease scans at loading rates of 0.1–10 pN s$^{-1}$, the force change during the stepwise bead height change (which occurs within a time window of <0.01 s, i.e., the temporal resolution of our setup) is negligible (≤0.001–0.1 pN). Consequently, the force-dependent stepwise bead height change during linear force-increase/decrease scans also equals to the extension change resulted from structural changes of the molecule.

## Theoretical calculation of the force-dependent transition step size

The force–extension curve of a folded dimer or domain is determined by the rigid rotational fluctuation of a characteristic rigid body with a length of $b$. The value of $b$ is the distance between the two force-attaching points and can be estimated from the PDB file of the folded domain or dimer. The force–extension curve of a rigid body can be described by the freely-jointed chain polymer model with a single segment:

$$x^{\text{rigid body}}(f) = b\left(\coth\left(\frac{fb}{k_{\text{B}}T}\right) - \frac{k_{\text{B}}T}{fb}\right) \qquad (1)$$

The force–extension curve of the unfolded state of a domain is determined by the force response of a flexible peptide chain. This curve can be described using the worm-like chain (WLC) polymer model with the Marko–Siggia formula[69], incorporating a bending persistence length $A \sim 0.8$ nm:

$$\frac{fA}{k_{\text{B}}T} = \frac{1}{4\left(1 - \frac{x^{\text{WLC}}(f)}{Nl_0}\right)^2} - \frac{1}{4} + \frac{x^{\text{WLC}}(f)}{Nl_0} \qquad (2)$$

where $l_0 = 0.38$ nm is the contour length per residue, and $N$ is the number of residues in the unfolded domain.

Hence, the force-dependent unfolding transition step size is the difference in extension of the domain before and after unfolding at the

transition force, given by:

$$\Delta x(f) = x^{\text{WLC}}(f) - x^{\text{rigid body}}(f) \qquad (3)$$

In the case of a dimer rupturing under shear–force geometry, where the rigid body length remains similar before and after rupture, the force-dependent step size corresponds to the extension difference of the looped long flexible linker before and after rupture at the transition force.

For the case of a dimer rupturing under unzip-force geometry, the force-dependent step size is the extension difference of the looped long flexible linker and the rigid body before and after rupture at the transition force.

For instance, an SR domain containing approximately 110 amino acid residues has a rigid body length of around 5 nm. A full SR rod domain, including the helical linker region, with about 480 amino acids has a rigid body length of approximately 24 nm. The rigid body length of a full SR rod dimer under unzip-force geometry is about 2 nm before rupture and $2 \times 24$ nm after rupture. The rigid body length of a full SR rod dimer under shear–force geometry is around 24 nm both before and after rupture. Based on these estimations, we can calculate the theoretical force-dependent transition step sizes (see Fig. S3).

## Effects of the flexible peptide chain linker

The total residue count in the flexible peptide chain linker is approximately 170 amino acids for the shear–force geometry dimer and approximately 218 amino acids for the unzip-force geometry dimer. Detailed sequence information is provided in Supplementary Note S1. The bending persistence of the flexible peptide chain linker is around 0.8 nm.

For the unzip-force geometry dimer, the end-to-end distance of the looped long flexible linker is estimated to be around 2 nm based on the structure. By utilizing the force–extension curve of the extended flexible linker with a worm-like-chain model, the looped long flexible linker can exert a force of less than 0.1 pN. This force is roughly 50 times smaller than the rupture force of the dimer (approximately 5 pN) under the unzip-force geometry in our experiments.

As for the shear–force geometry dimer, the end-to-end distance of the looped long flexible linker measures about 24, 18, and 12 nm for the full rod dimer, three SR dimer, and two SR dimer, respectively. According to the force–extension curve obtained using the worm-like-chain model for the extended flexible linker, the looped long flexible linker can exert a stretching force of less than 3.8 pN for the full rod dimer. This force is roughly 17 times smaller than the rupture force of the dimer (approximately 65 pN) under shear–force geometry in our experiments. Similarly, the looped long flexible linker can apply a stretching force of less than 2.6 pN for the three SR dimer, which is about 15 times smaller than the rupture force of the dimer (approximately 40 pN) under shear–force geometry in our experiments. In contrast, the looped long flexible linker can exert a stretching force of less than 1.5 pN for the two SR dimer, which is approximately 2 times smaller than the rupture force of the dimer (approximately 3 pN) under shear–force geometry in our experiments.

Together, these calculations suggest that due to the highly flexible nature of the long unstructured peptide chain, the extended flexible linker does not significantly disrupt the mechanical properties of the full rod dimer and the three SR dimers. Consequently, it does not impact the assessment of the mechanical stability of these dimers in our study. On the other hand, for the two SR dimer of $\alpha$-actinin 1 (the dimer formed by SR2–SR3:SR2–SR3), the effective stretching force experienced by the dimer is the combined result of the externally applied force and the stretching force applied by the linker.

## Bootstrap analysis

$N$ (>20) data points of the dimer's lifetime (dwell time of the dimer until rupture) were experimentally obtained for each force. To calculate the mean, standard deviation, and standard error of the dimer's rupture rate at each force, we employed bootstrap analysis on the experimental data. We generated $M = 20$ sets of data, each containing $N$ data points, by randomly selecting data from the original experimental dataset. To generate a new dataset with a size $N$, we randomly pick up one data point from the original data points pool, and then repeat this random pick-up procedure for $N$ times. Each time, the one data point is randomly picked up from the same original pool. In this way, the new dataset size does not necessarily to be smaller than the original dataset. Here we note that, a same original data point might be picked up more than once while some original data points might not be picked up in the new generated dataset. We repeated the above new dataset generation step for $M$ times. Consequently, we obtained $M$ sets of data, each comprising $N$ data points. Subsequently, for each dataset, we computed the time evolution of the rupture probability ($P(t)$) based on the $N$ data points within the dataset. The rupture probability data were then fitted to $P(t) = 1 - \exp(-k_{\text{rupture}} \times t)$ to determine the characteristic $k_{\text{rupture}}$ at this force for that dataset. The mean value, standard deviation, and standard error of $k_{\text{rupture}}$ were derived from a pool (with a size of M) of $k_{\text{rupture}}$ values. Similarly, the mean value, standard deviation, and standard error of the characteristic lifetime $\tau$ at this force were obtained from a pool (with a size of $M$) of $\tau = \frac{1}{k_{\text{rupture}}}$ values.

## Bell model analysis

The force-dependent rupture rate of the dimer can be described using the general Arrhenius Law as

$$k^{\text{Arrh}}(f) = k_0^{\text{Arrh}} \exp\left(\frac{\Delta\Phi_*^{\text{Arrh}}(f)}{k_B T}\right), \qquad (4)$$

where $\Delta\Phi_*^{\text{Arrh}}(f) = -\int_0^F \Delta_*^{\text{Arrh}}(f) df$, representing the free energy difference between the transition state and the initial state. Here, $\Delta_*^{\text{Arrh}}(f)$ stands for the force-dependent transition distance, which can be calculated based on the force-dependent extension difference between the transition state structure and the initial state structure. Given that both the transition state structure and the initial state structure for dimer rupture are rigid bodies, $\Delta_*^{\text{Arrh}}(f)$ can be approximated as a constant value $\Delta$. Consequently, the force-dependent rupture rates can be further approximated using the Bell model[70]:

$$k^{\text{Bell}}(f) = k_0 \exp\left(\frac{\Delta f}{k_B T}\right). \qquad (5)$$

By fitting the Bell model to experimentally determined force-dependent rupture rates, it becomes possible to extract key parameters such as the transition distance $\Delta$ and the zero-force rupture rate $k_0$ of a dimer. Utilizing bootstrap analysis, the mean, standard deviation, and standard error of both $\Delta$ and $k_0$ were obtained.

## Reporting summary

Further information on research design is available in the Nature Portfolio Reporting Summary linked to this article.

## Data availability

Key plasmids and protein constructs information is included in supplementary information. Key plasmids are available upon request from the corresponding authors. Source data are provided with this paper.

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

## Acknowledgements

The research is funded by the National Natural Science Foundation of China (NSFC Grant Nos. 32271367 and 12204389 to S.L., NSFC Grant No. 32301094 to M.Y., NSFC Grant No. 12174322 to H.C.), the Fundamental Research Funds for the Central Universities (20720220029 to S.L.), and the Ministry of Education under the Research Centres of Excellence programme (to J.Y.).

## Author contributions

S.L. and M.Y. conceived the study. S.L., M.Y., Y.Z., and J.D. designed and performed the experiments, and analyzed the data; S.L., M.Y., H.C., J.Y., Y.Z., J.D., Y.D., X.L., F.S., JQ.Y., and Y.W. interpreted the data; S.L., M.Y., H.C., and J.Y. wrote the paper with inputs from all authors.

## Competing interests

The authors declare no competing interests.
