## [Peer Review File · Nature Communications]

Multi-domain Interaction Mediated Strength-Building in Human α -Actinin Dimers Unveiled by Direct Single-molecule QuantificationREVIEWER COMMENTS

Reviewer #1 (Remarks to the Author):

In this manuscript, Zhang et al have used magnetic tweezers to study the mechanical stability of the alpha-actinin dimer and show that they are exceptionally stable under force despite the constituent interactions between the monomers being individually weak. The authors use a clever strategy (which they have employed in the past) in which the dimer is formed as part of a long molecular construct that includes flexible linkers as well as specific tags at its ends.

This study contains very interesting results on an important cytoskeleton-associated protein and is of broad interest. However, there are a few omissions and presumably correctable errors in the manuscript which if corrected will strengthen the paper.

Intro, para 4: The authors note that their "...findings reveal that these dimers exhibit ultra-high mechanical stability, boasting lifetimes > 100 seconds under shear-stretching geometry at forces within 40 pN". To support this claim in the Introduction, the authors ought to provide lifetimes of perhaps dimers of proteins where the lifetimes are much lower for similar forces (although they do some of this in the Discussion).

The authors have cited their previous paper (Chen et al, BiophysJ, 100, 517) wherein they have equated the force exerted as the $\Delta(M.B)$; however, Lipfert et al (BiophysJ, 96, 5040) have stated that an extra factor of $\frac{1}{2}$ should be included since the M of the superparamagnetic bead is induced by the external field. Can the authors state whether the factor $\frac{1}{2}$ should be included or not and their justification (regardless of whether this affects the actual reported force values in this paper)?

Dimer of alpha-actinin constructs with 4 SR domains does not exhibit multiple weaker states, unlike a dimer of alpha-actinin constructs with 3 SR domains – does this mean that there is some degree of cooperativity in the full length alpha-actinin dimerization? Can the authors comment on this?

Fig 1a: For all the different ΔH indicated in 1a, it may be beneficial to give the approximate expected values for them either in the figure or the caption – these values are only elsewhere in the text currently. This will enable more convenient reading of the plots in 1c.

Fig 1c: The gray arrows are supposed to indicate individual domain refolding; but the second gray arrow from the left, for instance, is not pointing to any stepdown in bead height (corresponding to a refolding) – are the arrows not placed correctly?

Results, para 2, last but 2 line: "...magenta curve in panel b" – did the authors mean Fig 1c, right panel? It is not clear which specific points in the plot refer to "unfolding of SR3-SR4, followed by the unfolding of SR1-SR2"

Fig 1g: How did the authors decide which data points to include in the Bell model fitting? For eg., how did they distinguish between the left most magenta and blue points as being in different groups?

Supplementary Text S9: The authors state that, for a given force, each of the M sets of data were made by randomly selecting N data points from an experimental dataset also of size N – did they not mean to call these both N? (since you are expected to randomly select a number of points that is less than the number of points in the experimental dataset?)

The relevance of including a DNA handle as in a configuration shown in Fig.S1 has not been substantially explained anywhere.

Fig 3d: Higher force levels shift the rupture probability curves to the right?! Mislabelled, perhaps?

Results, section 2 and 3: In the main text, the authors refer to Fig 2 instead of Fig 3 (and Fig 3 instead of Fig 2) leading to confusion

Discussion, line 5: Can the authors justify why a force-loading rate of 5 pN s⁻¹ is physiologically relevant?

Discussion, para 3, line 3: "...the rest of the alpha-actinin dimer decreases about 1000-folds." – something missing in this sentence

Results, section 2 heading makes no sense – perhaps remove 'Magnitudes'?

Discussion, para 3: "Since alpha-actinin monomer contains a single cryptic vinculin binding site buried in the folded SR4 domain" – the authors should cite their previous paper (Cell Rep. 2017, 21(10):2714-2723) here!

Results, section 4, para 3: The authors state that "...one of the SRs in the SR2-SR3-SR4 dimerizes with its partner, leaving the other two pairs undimerized in a weaker, non-native dimerized state (Fig. 4d, left panel)". However, this is not what the schematic of the dimer in fig 4d left panel shows

Fig 4h and 6e caption: hollow symbols are colored, rather than gray as stated

Supplementary text S6: A schematic diagram showing the forces and the torques will help readers visualize the torque rebalance subsequent to the force change referred to here

Methods, Line 10: Something is missing in this sentence.

Fig S1 caption and other places in the manuscript: streptavidin spelling

Reviewer #2 (Remarks to the Author):

In their study, Zhang et al study the mechanical strength of the antiparallel assembly of actinin dimers, a protein which is involved in cytoskeleton remodeling during cell-cell interaction through its four central rod domains. The authors designed an interconnected dimer through a flexible peptide linker and used magnetic tweezers to pull between one end of the molecule and a location in the middle of the molecule. They find that the tethered central rod domains can withstand forces of 40 pN for over 100 seconds and hypothesize that this molecule might form distinct dimerized functional states, mediated by weak, multivalent sub-domain interactions.

My main concern is that throughout the text the authors claim that they have shear-force stretching geometry. Such a geometry would imply that one dimer is pulled from one end and the other dimer from the opposing end. However what the experiment design produces here is that of four SR domains experiencing force in the presence of another four SR domains that do not. So basically what the authors are measuring is the mechanical stability of SR domains in the presence of their ligand, which in this case are the same SR domains in an antiparallel configuration. While it is an interesting question – how the mechanical stability of a protein changes when a ligand is bound – the molecular design does not reflect the process that takes place in vivo, where force is transmitted through the SR connections and both SR molecules are experiencing that force. So I don't think that the first statement in Results section is correct, which might be an issue, as the entire paper stands on this assumption. The experiment performed here also does not match the schematics in Fig 7, where the black and grey molecules are each under force.

The traces in Figure 1 seem to show a large unfolding step rather than four individual steps, which is >160 nm at forces over 60 pN. Figure 1a suggests that the authors use double stranded DNA to attach the middle of the protein to the bead, which also should have an overstretching transition at ~ 65 pN. The traces in Figure 1c either show no overstretching transition, or the measured step is the overstretching transition without any unfolding. Is the schematics in Figure 1a not representative of the traces in Figure 1c?

Histograms of refolding steps size and force are needed, as those coming from the traces in Figure 1c right. From that panel it looks like the steps are ~ 25 nm at $\sim 4-6$ pN. These values seem to contradict the prediction from Figure S2, which at these low forces predicts steps of 10-12 nm. How do the authors explain this contradiction?

Page 4 - "This led to rupture of the dimer along with the unfolding of SR3-SR4... as depicted in the magenta curve in panel b" (referring to Fig 1). But panel b in Fig 1 has just the force protocol. What are the authors talking about? Did the authors intended to add the length trace as well, but forgot?

How do the authors explain that the formation of the dimer results in a rather large step at force at ~ 3 pN? What is the mechanism behind this contraction "re-dimerization" step?

The schematics in Figure 1a and 2a look identical to me. The authors claim that data in Figure 1 was obtained in shear geometry, and in Figure 2 in unzipping geometry. Did the authors use the wrong schematics in Figure 2a? Also there is no "red arrow" in Figure 2b to indicate a rupture force, and the force histogram in Figure 2b bottom shows rather similar forces as the "shear" geometry, and not an average of 5 pN unzipping force, as claimed in the text. Similarly, Figure 2e should show a dimer with low stability at 4 pN, but has rather life time values at forces over 25 pN. Not sure what is going on here.

What are the force dependent rates that were measured from the hopping experiments in Figure 5c? Can they be used to estimate the binding/unbinding energy barriers?

Would have been interesting to investigate the nature of the "multivalent interaction" mechanism. Maybe some SMD simulations, paired with control experiments that could disrupt this interaction, would be a good path to follow.

Figure S8 shows a rather inhomogeneous distribution in the measured binding-unbinding kinetics, which the authors attribute to "weaker" and "strong" states. While such distributions are interesting and have been seen before for other proteins, this behavior does not seem to be addressed in the manuscript, except for "clearly distinguishable clusters". What could be the mechanism for these clusters?

Other comments:

It might be useful to include a representation from the crystal structure determined in ref. 14 as a panel in Figure 1 and reference it during the Introduction, where this protein is described.

It is not clear why the ancestral mutant brings to the table? How was it obtained? What sequence homogeneity does it have with the human actinin?

Text in figures is too small to be legible. Because figures were not rasterized, using Acrobat to read the pdf file is laggy.

Reviewer #3 (Remarks to the Author):

Overview: In this manuscript, the authors investigated mechanical stability of chimeric dimers of the cytoskeletal protein α -actinin isoforms. Dimers of α -actinin act in crosslinking actin to allow for cytoskeletal load bearing and assist in cellular mechanosensing and mechanotransduction. However, the mechanical stability of α -actinin dimers is not fully understood yet. They measured the effects of shear and unzip forces on α -actinin dimers. Magnetic tweezers were used to observe how force-induced rupturing of homo- and hetero-dimers of α -actinin affects their mechanical stabilities and lifetimes.

Overall, the experimental design to apply forces with magnetic beads was well thought out and the results could have significance in the field of cytoskeleton. Yet, the explanations of the main results and discussion needs to be clarified further to improve the manuscript.

Major Comments:

1. How do the forces used to load the dimers relate in comparison to physiological forces the actin network may encounter?
2. For the force jump-cycle procedure, how do the authors know the dimer was refolded properly after the dimer has been fully unfolded and then allowed to refold? Do the different dimers always refold the same way?
3. It would be helpful to clearly explain how unzip-force stretching is distinct from shear-force stretching. Is the bead attached to the n what direction is the bead moving to induce an unzip force that differs from a shear force?
4. How well do these SR based construct replicate the mechanical properties of the full-length α -actinin protein? Do the other domains known to be found in α -actinin not contribute to the mechanical strength?
5. The suggested kinetic modeling shown in Figure 7 is hard to grasp. Can the authors elaborate this suggested mechanism based on their results?

Minor Comments:

- The word "Multivalent" seems ambiguous.
- How similar is the sequence and structure of the *Entamoeba histolytica* α -actinin SR1 and SR2 compared to human α -actinin SRs?
- Figure 1 caption's title should include if the dimer was hetero or homo as was written for Figures 2-5.
- The authors mentioned their measurements being within a physiological force range. Are there any references to show what forces the α -actinin experience in the cell?

Main Changes and Point-by-point Responses to Reviewers' Comments

We thank all the reviewers for their thoughtful comments and helpful suggestions, which greatly helped us improve the manuscript. Please find below the list of main changes and the point-by-point responses to individual comments. Reviewers' comments are provided *verbatim* and our responses are in blue color within { }. The corresponding changes of the manuscript are noted at the end of each response. All the main changes listed below and other minor changes in the main text and supplementary text have been highlighted in red color.

Part I: List of Main Changes

Changes in Main Text:

- #1. Page 1, Title: the word "Multivalence" is revised to be "Multi-domain".
- #2. Page 1, author list: Two new co-authors, Jiaqing Ye and Yukai Wang, are added to acknowledge their contributions during the revision.
- #3. Page 2, Paragraph 3, Line 3-5: A sentence is revised to introduce the 90° end-to-end twisting of the SR pairs in the α -actinin dimer based on previous structural studies.
- #4. Page 2, Paragraph 4 & Page 3, Paragraphs 1-2: Two paragraphs are added to introduce the current understanding of physiological force, physiological time scale and physiological loading rate, as well as previous studies of mechanical stabilities of homo- and hetero- dimers formed by mechanosensitive proteins. Corresponding new references are cited (Change #14)
- #5. Page 4, Paragraph 1: The paragraph is revised to add more details on the experimental design and clarify the force-geometry of the experimental designs. The illustration panel in Figure 1 (panel a) is updated correspondingly (Change #15)
- #6. Page 4, Paragraph 2: The paragraph is revised to describe in more detail on the linear force-loading experimental procedures. The representative panel in Figure 1 (panel b) is updated correspondingly (Change #15)
- #7. Page 5, Paragraph 1, Line 3-7: Two sentences are added to clarify the large one step concurrent rupture & unfolding signal observed during linear force-increase scans.
- #8. Page 5, Paragraphs 3: A new result sub-section titled as "Force-dependent lifetime of α -actinin 1 Homo-dimer under shear-force Stretching Geometry" is splitted from previous results subsection 1. The corresponding figure of the sub-section is now Figure 2, which is splitted from previous Figure 1.
- #9. Page 5, Paragraph 3, Line 4-8: Two sentences are revised to clarify the force-jump experimental procedure.
- #10. Page 6, Paragraph 3, Line 6-13: Two sentences are revised to clarify the experimental design of the unzip-force stretching geometry.
- #11. Page 7, Paragraph 4, Line 4-5: One sentence is added to explain the potential molecular mechanism of the clustered mechanical groups of the dimer, which is further elaborated in the revised discussion section (Change #12).
- #12. Page 10, Paragraphs 3-5 & Page 11, Paragraphs 1-2: Four paragraphs are added to discuss the possible molecular mechanisms of the mechanical enhancement and cluster effect of the α -actinin dimer. The revised model includes a primary mechanism involving multi-domain pairs interaction kinetics, and additional possible factors including domain-pair re-orientation.
- #13. Page 12, Paragraph 1: Several sentence in the methods and materials section are revised to clarify the plasmids sub-cloning procedures.
- #14. Page 13-17: 29 new references are cited in the main text: Ref. 14, Ref. 21-22, Refs. 24-40, Refs. 42-47, Refs. 60-61 and Ref. 66.

Changes in Main Figures:

#15. Figure 1:

- i. Panel **a**: The illustration in panel a is revised to clarify the experimental design of the shear-force stretching geometry for α -actinin 1 homo-dimer.
- ii. Panel **a**: The structure example based on previous studies is added in the illustration panel.
- iii. Panel **a**: The theoretical calculated values of force-dependent rupture step size at some typical forces are labelled on the sketch.
- iv. Panel **b**: A representative time-bead height curve during force-increase scan and force-decrease scan cycles is added to replace the previous sketch of force-loading cycles. The corresponding events (rupture, unfolding, refolding, re-dimerization etc) are indicated with arrows in the panel.
- v. Panels **c-e**: The representative force-bead height curves of rupturing (during force-increase scans, left panel) and refolding/re-dimerization (during force-decrease scans, right panel), as well as the corresponding force-step size graph, and force distributions. The left panels of c-e and the right panel of c are the previous panels c-d. The right panels of d&e are new panels added during the revision. These two panels include the force-step size graph and force distributions of the SR domain refolding and dimer re-formation during force-decrease scans.
- vi. Previous panels e-g is now included in Revised Figure 2.
- vii. Figure captions are updated accordingly.

#16. Figure 2:

- i. Figure 2 is a new main figure, splitted from previous Figure 1.
- ii. Panel **a**: A representative time-bead height curve during force-jump scan cycles is added to clarify the force-jump experimental procedure for quantifying the force-dependent lifetime of the α -actinin 1 homo-dimer. The corresponding events (rupture, unfolding, refolding, re-dimerization etc) are indicated with arrows in the panel. Sketches of the dimer conformation is added on the panels. This panel is a replacement of previous experimental illustration of Figure 1e.
- iii. Panels **b&c** are previous Figure 1 **g&f**.
- iv. Figure captions are updated accordingly.

#17. Figure 3:

- i. The revised Figure 3 is previous Figure 2.
- ii. Panel **a**: The illustration is revised to clarify the experimental design of the unzip-stretching force geometry for α -actinin 1 homo-dimer. A theoretical calculation of the force-dependent step-size previously in panel b is now placed in panel a.
- iii. Panel **b**: arrows indicating different structural events (rupture, domain unfolding, refolding, re-dimerization) are added to increase the readability of the figure panel.
- iv. Panel **d**: the labelling of the three force values are updated correctly.
- v. Panel **c&e**: remain unchanged, except enlarged legends in the panels.
- vi. Figure captions are updated accordingly.

#18. Figure 4:

- i. The revised Figure 4 is previous Figure 3.
- i. Panel **a**: The illustration is revised to clarify the experimental design of the shear-stretching force geometry for α -actinin 4 homo-dimer.
- ii. Panels **b&c**: The representative force-bead height curves of rupturing (during force-increase scans, left panel) and refolding/re-dimerization (during force-decrease scans, right panel), as well as the corresponding force-step size graph, and force distributions. The panels of **b** and the top panel of **c** are the previous

panels **b&c**. The middle and bottom panels of **c** are new panels added during the revision. These two panels include the force-step size graph and force distributions of the SR domain refolding and dimer re-formation during force-decrease scans for α -actinin 4 homo-dimer.

- iii. Panel **d&e**: remain unchanged, except enlarged legends in the panels.
- iv. Figure captions are updated accordingly.

#19. Figure 5:

- i. The revised Figure 5 is previous Figure 4.
- ii. Panels **a-d**: The illustrations in the left panels are updated with additional words descriptions for the potential conformational changes corresponding to the data shown in the right panels.
- iii. Previous panel e is now revised as a Supplementary Figure S11.
- iv. Panel **e** is updated from previous panel f, with theoretical calculations of force-dependent step sizes of the potential conformational changes during the rupturing of the dimer.
- v. Panel **f** is the previous bottom panel of panel f.
- vi. Panel **g&h**: remain unchanged, except enlarged legends in the panels.
- vii. Figure captions are updated accordingly.

#20. Figure 6:

- i. The revised Figure 6 is previous Figure 5.
- ii. Panels **a&b**: The illustration of the design and the representative curves are rearranged to increase the readability without data content changes.
- iii. Panel **c**: To increase the readability of the figure, previous 200 seconds time-bead height curves of the 2-SR dimer is now updated with a 20 second zoom-in figure associated with the distributions of the two states (dimerized and un-dimerized states) on the right.
- iv. Panels **d-f**: three new panels are added to include the data analysis of the dimerization and un-dimerization dynamics of the dimer and the corresponding forces.
- v. Figure captions are updated accordingly.

#21. Figure 7:

- i. The revised Figure 7 is previous Figure 6.
- ii. The panel **a**: the illustration is updated to increase the readability of the experimental design.
- iii. The panel **b**: arrows indicating the dimer rupture events are added to increase the readability of the figure panel.
- iv. Panels **c-e**: remain unchanged.
- v. Figure captions are updated accordingly.

#22. Figure 8:

- i. the model is revised based on previous Fig. 7. The illustration is updated to increase the readability of the model.
- ii. Figure captions are updated accordingly.

Changes in Supplementary Texts:

- #23. Supplementary Text S9: The description of the bootstrap analysis is revised to clarify the data point sampling during the analysis.

Changes in Supplementary Figures:

- #24. Supplementary **Figure S1**: The Figure caption is revised to include the reasoning of using the DNA handle in the stretching configuration.
- #25. Supplementary **Figure S2**: A new supplementary Figure S2 is added to illustrate the bead rotation due to torque rebalance in magnetic tweezers experiments.
- #26. Supplementary **Figure S4**: a new supplementary Figure S4 is added to include some more example force-bead height curves of the α -actinin 1 homo-dimer under shear-force geometry during force-increase scans.
- #27. Supplementary **Figure S5**: it is previous Figure S3. The Figure caption is revised to include descriptions of the results obtained with or without DNA handle in the single-molecule stretching configurations.
- #28. Supplementary **Figure S9**: it is previous Figure S7. The illustration in panel **a** is revised to clarify the experimental design of the shear-stretching force geometry for α -actinin 1&4 hetero-dimer. The panels b&c are now re-arranged with the representative force-bead height curves of rupturing (during force-increase scans, left panel) and refolding/re-dimerization (during force-decrease scans, right panel), as well as the corresponding force-step size graph, and force distributions. The panels of b and the top panel of c are the previous panels b&c. The middle and bottom panels of c are new panels added during the revision. These two panels include the force-step size graph and force distributions of the SR domain refolding and dimer re-formation during force-decrease scans for α -actinin 1&4 hetero-dimer. The other panels remained unchanged. The Figure captions are updated accordingly.
- #29. Supplementary **Figure S11**: it is a new supplementary Figure. The panels b&d are from panel e of previous main Figure 4. The panel c is updated with theoretical calculations of force-dependent step sizes of the domains unfolding.
- #30. The contents of other supplementary Figures remain unchanged while the numberings are updated due to the insertion of new supplementary figures. The citations of the supplementary figures in the main text are updated accordingly.

Part II: Point-by-point Responses to Reviewers' Comments

Comments #1:

Reviewer #1 (Remarks to the Author):

In this manuscript, Zhang et al have used magnetic tweezers to study the mechanical stability of the alpha-actinin dimer and show that they are exceptionally stable under force despite the constituent interactions between the monomers being individually weak. The authors use a clever strategy (which they have employed in the past) in which the dimer is formed as part of a long molecular construct that includes flexible linkers as well as specific tags at its ends.

This study contains very interesting results on an important cytoskeleton-associated protein and is of broad interest. However, there are a few omissions and presumably correctable errors in the manuscript which if corrected will strengthen the paper.

Responses #1:

{

We thank the reviewer for his supportive and constructive comments and suggestions. Accordingly, we have made revisions throughout the manuscript.

}

Comments #2:

Intro, para 4: The authors note that their "...findings reveal that these dimers exhibit ultra-high mechanical stability, boasting lifetimes > 100 seconds under shear-stretching geometry at forces within 40 pN". To support this claim in the Introduction, the authors ought to provide lifetimes of perhaps dimers of proteins where the lifetimes are much lower for similar forces (although they do some of this in the Discussion).

Responses #2:

{

We acknowledge the reviewer for providing a constructive suggestion for introduction revision. Accordingly, we added a new paragraph to include several examples of previously reported force-dependent lifetimes of some key mechanosensitive homo- or hetero- dimers.

Change List # 4.

Also see in the Main Text: Page 2, Paragraph 4 & Page 3, Paragraphs 1-2.

}

Comments #3:

The authors have cited their previous paper (Chen et al, BiophysJ, 100, 517) wherein they have equated the force exerted as the del (M.B); however, Lipfert et al (BiophysJ, 96, 5040) have stated that an extra factor of 1/2 should be included since the M of the superparamagnetic bead is induced by the external field. Can the authors state whether the factor 1/2 should be included or not and their justification (regardless of whether this affects the actual reported force values in this paper)?

Responses #3:

{

The strict equation of the potential energy of a paramagnetic bead in magnetic field B is

$$U = - \int_0^B M(B)dB,$$

where M is the magnetic moment of paramagnetic bead which is a function of B , and B is the magnetic field. At small field B , M is a linear function of B , and the integral gives $U = -\frac{1}{2}MB$. While at field which is much larger than the saturation field, U is approximately equal to MB .

Force is the minus gradient of the potential energy:

$$F = -\nabla U = M(B) \frac{d}{dz} B,$$

which is always valid in any magnetic field. There is no factor $\frac{1}{2}$ in this formula of force.

As both the magnetic field and magnetic moment are difficult to measure, the force calibration in practical does not depend on this equation. The force values in the paper are not affected. In typical magnetic tweezers setup, the force is calibrated based on the thermal fluctuation of the position of the molecule-tethered bead and the relation between the force and the magnets-to-bead distance.

}

Comments #4:

Dimer of alpha-actinin constructs with 4 SR domains does not exhibit multiple weaker states, unlike a dimer of alpha-actinin constructs with 3 SR domains – does this mean that there is some degree of cooperativity in the full length alpha-actinin dimerization? Can the authors comment on this?

Responses #4:

{

The reviewer raised a very interesting and important point regarding the different behaviours of the 4 SR domain dimer and 3 SR domain dimer. We agree with the reviewer that it may suggest certain degree of cooperativity in the SR domains dimerization.

For the 4 SR domain dimer (i.e., the full SR rod dimer), in our experiments, we primarily observe a strong mechanical lifetime group for most of the molecules stretched (major rupture force peak in Fig. 1e, and blue data in Fig. 2c). Weaker mechanical lifetime groups were only observed for a few cases (for instance, magenta and orange data in Fig. 2c). In contrast, the 3 SR domain dimer shows at least four different mechanical lifetime groups, from weak to strong (Fig. 5). Interestingly, for the 2 SR domain dimer, we could only observe one weak mechanical lifetime group within our experimental time scale (Fig. 6). In addition, we could not observe 1 SR domain pair dimer (Fig. S12). Furthermore, the mechanical stability dramatically increases from ~3 pN for 2 SR domain pair dimer to ~40 pN for 3 SR domain pair dimer (strong group), and ~67 pN for 4 SR domain pair dimer (strong group), in a non-linear manner. These results may suggest cooperativity of the multi-domain dimerization involving the cooperative pairing of the domain pairs, which will be explained below.

We reason that the cooperativity may be resulted from one primary factor and potentially several additional factors. The primary factor is the multivalent interaction of the multi-domain dimerization. Since the dissociated domains remain in proximity due to the remaining of dimerized domain pairs, rapid re-association of the dissociated domain becomes possible, leading to stabilization through multivalency. This factor could explain the dramatic increases in mechanical stability (from weak to strong mechanical lifetimes) as domain-pair number increases. This was the model proposed in Fig. 7 in our previous submission, and revised as Fig. 8 in the revision.

The additional factor may involve the re-orientation of the domain pairs during forming the native rod dimer. The structural studies of the α -actinin dimer have revealed that the anti-parallel SRs in the rod are organized with certain degree of re-orientation along the rod axis,

resulting in twisting about 90° from one end to the other end. This suggest that the SR pairs should be not only paired, but also re-oriented with certain degree in the native dimerization state.

The SR rod monomer contains four SR domains linked by a short helical linker. The linker between two neighbouring SR domains is a short rigid helix extended from the C-terminal helix of the former SR or the N-terminal helix of the later SR (Ref. 15). When the SR rods form dimers, the SR domains in the rod not only need to be paired, but also need to be re-oriented to suit into the natively 90° twisted conformation. Due to the rigidity of the helical linker and the domains, the domain-pairs may be trapped in certain intermediate conformational state where some of the domains pair and re-orient into the native states, while some of the domains pair in non-native states. Energy barriers between the states may lead to the clustered distinct mechanical stability group. On the other hand, the domain-pair re-orientation also implies that natively re-oriented domain-pairs may facilitate their neighbouring domain-pair's re-orientation, leading to cooperativity of paring and re-orientation of the domains, which in turn enhances the mechanical stability of the dimer.

In the 4 SR domain dimer (i.e., the full SR rod dimer), due to the cooperativity, it primarily stays in the natively paired and re-oriented states, leading to primarily the strong mechanical group of the dimer. In the 3 SR domain dimer, such cooperativity effect may be weakened, leading to certain degree of freedom of the domain-pair's re-orientation state, which results in multiple states of the dimer associated with distinct mechanical lifetimes. For instance, two of the SR pairs in the native orientation, while another pair stays in un-native orientation. Consistently, in the 2-SR dimer of human α -actinin, such cooperativity effect on the re-orientation further weakened, resulting in one weak mechanical lifetime group observed experimentally.

We thank the reviewer for the stimulating comments, we have revised our model and included a new paragraph of discussion on this aspect in the revised manuscript.

Change List # 12.

Also see in the Main Text: Page 10, Paragraphs 3-5 & Page 11, Paragraphs 1-2.

}

Comments #5:

Fig 1a: For all the different ΔH indicated in 1a, it may be beneficial to give the approximate expected values for them either in the figure or the caption – these values are only elsewhere in the text currently. This will enable more convenient reading of the plots in 1c.

Responses #5:

{

Thank the reviewer's suggestion. We have added the expected values for the force-dependent step sizes in figure panels in the revised manuscript.

Change List # 15.

Also see in the Main Text: Page 18, Figure 1 and its captions.

}

Comments #6:

Fig 1c: The gray arrows are supposed to indicate individual domain refolding; but the second gray arrow from the left, for instance, is not pointing to any stepdown in bead height (corresponding to a refolding) – are the arrows not placed correctly?

Responses #6:

{

Thank the reviewer's reminder. The arrows were accidentally horizontally shifted during figure panel preparation using adobe illustrator. Sorry for the mistake and inconvenience caused. We have corrected it in the revised figures.

Change List # 15.

Also see in the Main Text: Page 18, Figure 1 and its captions.

}

Comments #7:

Results, para 2, last but 2 line: "...magenta curve in panel b" – did the authors mean Fig 1c, right panel? It is not clear which specific points in the plot refer to "unfolding of SR3-SR4, followed by the unfolding of SR1-SR2"

Responses #7:

{

Thank the reviewer's reminder again. Yes, it should be Fig. 1c. In our previous submission, a panel b (force-loading procedure sketch) was added in the Figure but forgot to update the description in the panel. Sorry for the mistake. We have updated the main Figures and supplementary figures during revision. The mentioned data is now included as supplementary Figure S4 in the revised manuscript.

Change List # 26.

Also see in the Main Text: Page 5, Paragraph 1, last sentence.

}

Comments #8:

Fig 1g: How did the authors decide which data points to include in the Bell model fitting? For eg., how did they distinguish between the left most magenta and blue points as being in different groups?

Responses #8:

{

During our experiments and data analysis, we found that the distinct lifetime distribution occurs in a clustered mode. As the reviewer pointed out in Comment #4, it is particularly obvious for the three SRs dimer (SR2-SR3-SR4: SR1-SR2-SR3). For the four SRs full rod dimer, we mainly observe the strong lifetime group (the blue data in previous Fig. 1g, now revised Fig. 2c) during the force-jump measurement within our experimental time scale. For previous Fig. 1g (now revised Fig. 2c) specifically, the magenta data were obtained from one molecule and the orange data were obtained from another molecule. We reason that these two specific molecules were in weaker dimerization states when it was stretched and maintained in the mode during our measurement. Hence, data from these two molecules were excluded for Bell model fitting for the strong lifetime group.

Change List # 15&16.

Also see in the Main Text: Page 18, Figure 1 and its captions & Page 19, Figure 2 and its captions.

}

Comments #9:

Supplementary Text S9: The authors state that, for a given force, each of the M sets of data were made by randomly selecting N data points from an experimental dataset also of size N

– did they not mean to call these both N? (since you are expected to randomly select a number of points that is less than the number of points in the experimental dataset?)

Responses #9:

{

In the bootstrap analysis, to generate a new dataset with a size N, we randomly pick up one data point from the original data points pool, and then repeat this random pick-up procedure for N times. Each time, the one data point is randomly picked up from the same original pool. In this way, the new dataset size does not necessarily to be smaller than the original dataset. Here we note that, a same original data point might be picked up more than once while some original data points might not be picked up in the new generated dataset.

We have revised the description of the bootstrap analysis method section in the supplementary text to clarify this aspect.

Change List #23.

Also see in Supplementary Text S9.

}

Comments #10:

The relevance of including a DNA handle as in a configuration shown in Fig.S1 has not been substantially explained anywhere.

Responses #10:

{

Thanks for the reviewer’s reminder. The DNA handle illustrated in the Fig. S1 acts as an initial additional positive control to show the specificity of the target signal (previous Fig. S3b, which is now as revised Fig. S5b). A specific characteristic DNA overstretching signal at ~ 65 pN helps to ensure that a specific single target molecule was stretched at the initial experimental test period when probing the force-dependent dynamics of a protein/protein complex. Signals observed other than the characteristic DNA overstretching signal should be resulted from the target protein dynamics. In the meantime, the characteristic SR refolding signals at 3-6 pN forces could also act as positive control.

Since the unexpected high mechanical stability of SR rod dimer that ruptures at > 60 pN, similar with the force range of DNA overstretching, we then removed the DNA handle in the experiments. All data for the figures except the previous Fig. S3b was obtained by a configuration without DNA handle.

We have clarified this point in the figure captions of the revised Fig. S1 and Fig. S5b. In addition, we also updated sketch of Fig. 1a to increase the readability of the experimental design.

Change List #15, 24 & 27

}

Comments #11:

Fig 3d: Higher force levels shift the rupture probability curves to the right?! Mislabeled, perhaps?

Responses #11:

{

Thank the reviewer's kind reminder. The legend in previous Fig. 3d was mislabelled. It should be labelled in a reverse order. We have corrected the mislabelling in the revised manuscript.

Change #17

Also see in the Main Text: Page 20, Figure 3 and its captions.

}

Comments #12:

Results, section 2 and 3: In the main text, the authors refer to Fig 2 instead of Fig 3 (and Fig 3 instead of Fig 2) leading to confusion

Responses #12:

{

Thank the reviewer's kind reminder. We apologize for the mislabelling in the first submission. We did a major reorganization of the figure panels before the previous submission, which led to mislabelling of the figures. We have corrected all the mislabelling of the figures in our revised manuscript. In addition, since a new Figure 2 is added in the revision, the previous Figures 2&3 are now Figures 3&4 in the revision.

Again, we apologize for these mistakes.

Change List #17&18

Also see in the Main Text: Page 20, Figure 3 and its captions & Page 21, Figure 4 and its captions.

}

Comments #13:

Discussion, line 5: Can the authors justify why a force-loading rate of 5 pN s^{-1} is physiologically relevant?

Responses #13:

{

The reviewer pointed out a very important aspect of the physiologically relevant force loading rate. To estimate a physiological relevant force loading rate, two main parameters should be estimated: the force range and the force-bearing time scale of a molecule *in vivo*. The physiological force applied on a single force-bearing protein are expected to be a few pN to tens of pN. The primary source of the force is the actomyosin contraction. It has been previously estimated that a single myosin could generate about 3 pN force (Ref. 39). A typical myosin fiber contains about 10-12 myosin heads, which leads to about 3-36 pN forces depending on how many myosin heads stay on the actin simultaneously (Ref. 40). Such a few to tens of pN force range have been evidenced by cellular FRET-based force-sensor measurement for many classic force bearing proteins, such as talin, vinculin, catenins, cadherins etc (Refs. 24-38). The force-bearing time scale depends on the interaction time scale of the force-bearing proteins. The interaction time scale for these proteins are typically within seconds to a few hundreds of seconds estimated based on FRAP (Fluorescence recovery after photobleaching) experiments (Refs. 24,31,41). Altogether, the physiological relevant force loading rate hence could be roughly estimated to be in the scale of pN/s. Consistent with this rough estimation, a recent study by Ha Lab estimated that the integrin loading rate is ranged from 0.5 pN/s to 4 pN/s, using a novel FRET-based DNA force-loading sensor (Ref. 42).

A new paragraph of introduction on the physiological force, physiological force-dependent interaction time scale, and physiological loading rate is added in the revised manuscript. Corresponding new references are cited.

Change List # 4.

Also see in the Main Text: Page 2, Paragraph 4 & Page 3, Paragraphs 1-2.

}

Comments #14:

Discussion, para 3, line 3: "...the rest of the alpha-actinin dimer decreases about 1000-folds."
– something missing in this sentence

Responses #14:

{

Sorry for the confusion. The previous sentence "When one of the SR1:SR4 is un-dimerized, the rest of the alpha-actinin dimer decreases about 1000-folds" was intended to describe the fact that when one SR1:SR4 pair is in un-dimerized state, the lifetime of the remaining dimerized pairs of the dimer (i.e., a dimer with SR2-SR3-SR4:SR1-SR2-SR3) decreases about 1000-folds compared to that of the full dimer. We have rephrased the sentence to make it clearer.

See Change in the Main Text: Page 9, Paragraph 4, Lines 3-4.

}

Comments #15:

Results, section 2 heading makes no sense – perhaps remove 'Magnitudes'?

Responses #15:

{

Thanks for the suggestion, we have removed the word accordingly.

}

Comments #16:

Discussion, para 3: "Since alpha-actinin monomer contains a single cryptic vinculin binding site buried in the folded SR4 domain" – the authors should cite their previous paper (Cell Rep. 2017, 21(10):2714-2723) here!

Responses #17:

{

Thanks for the reminder. We have included the reference accordingly.

}

Comments #17:

Results, section 4, para 3: The authors state that "...one of the SRs in the SR2-SR3-SR4 dimerizes with its partner, leaving the other two pairs undimerized in a weaker, non-native dimerized state (Fig. 4d, left panel)". However, this is not what the schematic of the dimer in fig 4d left panel shows Fig 4h and 6e caption: hollow symbols are colored, rather than gray as stated

Responses #17:

{

Thanks for the reviewer's reminder. In previous Fig. 4d (now revised Fig. 5d), the sketch in the left panel was intended to show a potential scenario that the dimer formed mainly by one of the SR pairs, while the other two SR pairs associated together in non-native state, which leads to the weak mechanical stability of the dimer that could rupture at forces lower than the domain unfolding forces (data present in the right panel of previous Fig. 4d, now Fig. 5d in the revised manuscript). We have revised the sketch to better present such a possible scenario in the revised manuscript.

The typo of the figure caption of previous Fig 4h and 6e has been corrected in the revised manuscript.

Change List #19.

Also see in the Main Text: Page 22, Figure 5 and its captions.

}

Comments #18:

Supplementary text S6: A schematic diagram showing the forces and the torques will help readers visualize the torque rebalance subsequent to the force change referred to here.

Responses #18:

{

Thanks for the reviewer's suggestion. We have included a schematic diagram of the forces and torque rebalance as a new supplementary Figure S2 and quoted it in the supplementary text S6.

Change List #25

}

Comments #19:

Methods, Line 10: Something is missing in this sentence.

Responses #19:

{

We have revised the sentences in the methods section to improve the readability of the sections.

Change List #13

Also see Page 11, Last Paragraph in the Main Text.

}

Comments #20:

Fig S1 caption and other places in the manuscript: streptavidin spelling

Responses #20:

{

Thanks for the reviewer's kind reminder. We have corrected the spelling and also tried our best to correct the grammar error and typos throughout the manuscript.

}

Comments #21:

Reviewer #2 (Remarks to the Author):

In their study, Zhang et al study the mechanical strength of the antiparallel assembly of actinin dimers, a protein which is involved in cytoskeleton remodeling during cell-cell interaction through its four central rod domains. The authors designed an interconnected dimer through a flexible peptide linker and used magnetic tweezers to pull between one end of the molecule and a location in the middle of the molecule. They find that the tethered central rod domains can withstand forces of 40 pN for over 100 seconds and hypothesize that this molecule might form distinct dimerized functional states, mediated by weak, multivalent sub-domain interactions.

My main concern is that throughout the text the authors claim that they have shear-force stretching geometry. Such a geometry would imply that one dimer is pulled from one end and the other dimer from the opposing end. However what the experiment design produces here is that of four SR domains experiencing force in the presence of another four SR domains that do not. So basically what the authors are measuring is the mechanical stability of SR domains in the presence of their ligand, which in this case are the same SR domains in an antiparallel configuration. While it is an interesting question – how the mechanical stability of a protein changes when a ligand is bound – the molecular design does not reflect the process that takes place in vivo, where force is transmitted through the SR connections and both SR molecules are experiencing that force. So I don't think that the first statement in Results section is correct, which might be an issue, as the entire paper stands on this assumption. The experiment performed here also does not match the schematics in Fig 7, where the black and grey molecules are each under force.

Responses #21:

{

We acknowledge the reviewer's constructive comments. The main concern of the reviewer is about the shear-force design of the experiments. We apologize for the confusion caused due to the previous illustration in Fig. 1a. We also thank the reviewer for raising this point so that we can revise our manuscript to improve the readability.

When the dimer interacts with F-actins, it is stretched via its two N-terminus ABDs, leading to both of the α -actinin rods in the dimer experience forces from their N-termini. Since the two SR rods in the dimer is anti-parallel, the force applied at the two N-termini is in a shear-force geometry. To mimic this scenario, we designed a chimeric protein construct from which the two N-termini of two SR rods are stretched under force.

In the designed chimeric protein construct, it contains (from N-terminus to C-terminus): a SpyTag, an α -actinin SR rod, a long flexible linker, a biotinylated AviTag, and another α -actinin SR rod. The chimeric construct was specifically tethered to a SpyCatcher-coated coverslip surface via SpyTag, and to a streptavidin-coated paramagnetic bead surface via biotinylated AviTag. The two force-attaching points of the construct are SpyTag at the N-terminus of the first α -actinin SR rod and the biotinylated AviTag at the N-terminus of the second α -actinin SR rod. Hence, when the SR rod dimer formed at low forces, the two N-termini of the two SR rods are stretched under force in a shear-force geometry. When the dimer is ruptured, only the first α -actinin rod (between the SpyTag and biotinylated AviTag) is under force, the second α -actinin rod (the domains after biotinylated AviTag) is free of force. Once the dimer re-formed, both rods again experience external forces from the two N-termini in a shear-force geometry.

The schematics in Fig. 7 is a modelled dimer where the monomer in the dimer is anti-parallelly paired, and experience external forces through the two N-termini, i.e., shear-force geometry. As explained above, the experiments designed in the study match the modelled dimer in schematics in Fig. 7.

A unique feature of the design is the long flexible linker that connect the two rods. The long flexible linker provides several advantages while it does not affect the quantification of the mechanical stability of the dimer: (i) It serves as an amplifier of the rupturing signal through a large extension jump upon rupture of the dimer. (ii) It allows rapid re-formation of the ruptured dimer at lower forces by keeping the freed monomer in vicinity, which largely increases the experimental throughput. (iii) It provides a molecular signature to eliminate any contamination from nonspecific interactions. (iv) Being a highly flexible unstructured peptide chain that has a bending persistence of ~ 0.8 nm and a contour length of ~ 0.38 nm per residue, when the dimer forms, the looped flexible linker does not exert significant force to the dimer. Hence, such a flexible long linker design could be implemented for probing the mechanical stability of dimers.

The force-attaching points of the construct could be controlled by designing the position of the tags. For instance, if we place the two tags (SpyTag and biotinylated AviTag) at the N-terminus of each rod, the force will be applied via the two N-termini of monomers in the dimer (shear-force geometry), if we place one tag at the N-terminus of the first rod, and the other tag at the C-terminus of the second rod, the force will be applied via the N-terminus of the first rod and the C-terminus of the second rod (unzip-force geometry).

We believe that the confusion was caused by that we did not clearly label the force-attaching points of the design in our previous Fig. 1a. In our revision, we have updated the Fig. 1a with clearly labelled force-attaching point of the construct with black upward arrows. We have also revised the first paragraph of the revised Results section, to include more detailed descriptions of the experimental design.

Change List #5, and 15.

Also see in the Main Text: Page 4, Paragraph 1 & Page 18, Figure 1 and its captions.

}

Comments #22:

The traces in Figure 1 seem to show a large unfolding step rather than four individual steps, which is >160 nm at forces over 60 pN. Figure 1a suggests that the authors use double stranded DNA to attach the middle of the protein to the bead, which also should have an overstretching transition at ~ 65 pN. The traces in Figure 1c either show no overstretching transition, or the measured step is the overstretching transition without any unfolding. Is the schematics in Figure 1a not representative of the traces in Figure 1c?

Responses #22:

{

We thank the reviewer's constructive comments and sorry for the confusion caused here. The data in Figure 1 (as well as all the figures in the main text) were obtained in a configuration without DNA handle (see updated Fig. 1a). In Fig. 1, the large step >160 nm at forces over 60 pN was resulted from the rupture of the dimer. When the dimer ruptures, it releases the looped long flexible linker. In addition, since the SR domain monomers unfold at forces of ~ 15 pN and ~ 28 pN, respectively (Ref. 41). When the dimer ruptures force at ~ 60 pN, the freed SR domains would unfold immediately together with the release of the long linker, leading to the concurrent rupture and unfolding as a single large step.

The force-dependent step sizes of the concurrent rupture and unfolding is consistent with theoretical calculation assuming concurrent the long flexible linker releasing with four SR domain unfolding (Fig. 1d, circles) or with three SR domain unfolding (Fig. 1d, triangles). In the later scenario, one SR domain unfolding was observed prior the concurrent rupture and domain unfolding (example, Fig. 1C, orange curve). In addition, in each scan cycle, we ensure the refolding of the four SR domains and re-dimerization of the two rod by force-decreasing scan with slow force loading rate (-0.1 pN/s), which provides clear four refolding signals and one re-dimerization signals.

Next, we explain the cause of the confusion on DNA handle. We performed the experiments with two configurations: with or without dsDNA handle. For the experiments with dsDNA handle, we observed both the rupture steps (>160 nm) at forces over 60 pN, and the dsDNA overstretching step indicated by its characteristic elongation behaviour at ~ 65 pN (Fig. S5). The DNA handle was included at the initial experimental test as it provides a specific characteristic DNA overstretching signal at ~ 65 pN to ensure that a specific single target molecule was stretched. Since the unexpected high mechanical stability of SR rod dimer that ruptures at > 60 pN, similar with the force range of DNA overstretching, we then removed the DNA handle in the experiments. All data for the figures except the previous Fig. S3b (now Fig. S5b) was obtained by a configuration without DNA handle.

We are sorry for forgetting to update the illustration for in the previous submission. We have updated the Fig. 1a in the revised manuscript. We also updated the corresponding figure captions. The traces with dsDNA handle was included as supplementary Figure S5b..

Change List #6, 7, 15, 24 and 27.

}

Comments #23:

Histograms of refolding steps size and force are needed, as those coming from the traces in Figure 1c right. From that panel it looks like the steps are ~25 nm at ~4-6 pN. These values seem to contradict the prediction from Figure S2, which at these low forces predicts steps of 10-12 nm. How do the authors explain this contradiction?

Responses #23:

{

We thank for the reviewer's suggestion to include the histograms of refolding steps and force. Accordingly, the force—step sizes graph of the SR domain refolding as well as the rod re-dimerization are added as new panel (1d, right panel) in revised Figure 1. We also added the normalized distributions of refolding forces and re-dimerization forces as new Fig. 1e right panel.

In addition to the α -actinin 1 homo-dimer in Figure 1 as the reviewer mentioned, we also plotted the force—step sizes graphs and the normalized force distributions of the α -actinin 4 homo-dimer as well as α -actinin 1&4 hetero-dimers in Figure 4c and Figure S9, respectively.

As shown in Figure 1d, the re-folding step sizes of individual SRs were ~10-12 nm at ~ 4-6 pN, consistent with the theoretically calculated force-step sizes (plotted as dashed line in the panel). In addition, the force-dependent step size of re-dimerization is also in consistent with the theoretical values. Here we note that, for a few data points of domain folding (gray circles in Fig. 1d, right panel) where the step size is about twice of the other values was due to refolding of two SR domains within a time scale narrower than our experimental time resolution.

A few outlier data points of the re-dimerization step sizes (dark gray circles in Fig. 1d, right panel) were due to re-dimerization immediately after the refolding of the domains within our experimental time resolution.

The reason that it seems to be larger steps in previous Fig. 1c (now as revised Fig. 1c, right panel) was due to the height-to-width ratio of the figure panel in order to place all the figure panels in one figure.

Change List #15, 18 and 28.

}

Comments #24:

Page 4 - "This led to rupture of the dimer along with the unfolding of SR3-SR4... as depicted in the magenta curve in panel b" (referring to Fig 1). But panel b in Fig 1 has just the force protocol. What are the authors talking about? Did the authors intended to add the length trace as well, but forgot?

Responses #24:

{

We are very sorry for the confusion caused due to our mistake that the previous panel b was updated with a force protocol while did not update the descriptions in the text. We have included the traces as a new supplementary Figure S4.

Change List # 26.

Also see in the Main Text: Page 5, Paragraph 1, last sentence.

}

Comments #25:

How do the authors explain that the formation of the dimer results in a rather large step at force at ~3 pN? What is the mechanism behind this contraction "re-dimerization" step?

Responses #25:

{

The large stepwise decrease of the extension during dimer formation was resulted from the looping the long flexible linker when the dimer formed. When the dimer ruptures, the long flexible linker is under forces and has a force-dependent extension. When the dimer forms, the long flexible linker is looped within the dimer and is not under forces. The extension differences between the looped and released long flexible linker leads to the large step decrease when the dimer forms. Such a long flexible linker acts as a specific indicator of rupturing and re-dimerization. As the reviewer suggested in Comment #23, we analysed the force-dependent step sizes of both domains refolding and re-dimerization during the force-decrease scans and plotted the experimental data with the theoretical calculations based on polymer physics models. The experimental data and the theoretical calculations are well consistent with each other (the revised Figure 1 c-e, Figure 4 b&c, Figure S9 b&c).

Change List #15, 18 and 28.

}

Comments #26:

The schematics in Figure 1a and 2a look identical to me. The authors claim that data in Figure 1 was obtained in shear geometry, and in Figure 2 in unzipping geometry. Did the authors use the wrong schematics in Figure 2a? Also there is no “red arrow” in Figure 2b to indicate a rupture force, and the force histogram in Figure 2b bottom shows rather similar forces as the “shear” geometry, and not an average of 5 pN unzipping force, as claimed in the text. Similarly, Figure 2e should show a dimer with low stability at 4 pN, but has rather life time values at forces over 25 pN. Not sure what is going on here.

Responses #26:

{

We again are very sorry for the confusion caused. The Figure 2 and its Figure caption in the first submission were for actinin 4 dimer, while the Figure 3 and its figure caption were for actinin 1 dimer in unzipping geometry. We did a major figure panel re-organization before previous submission to reduce figure numbers. However, we missed out some updating of the text descriptions in the results section and figure captions.

We have corrected all the wrong figure labelling in the revised manuscript. The previous Figures 2 and 3 are now revised Figures 3 and 4, respectively.

Change List #17 and 18.

}

Comments #27:

What are the force dependent rates that were measured from the hopping experiments in Figure 5c? Can they be used to estimate the binding/unbinding energy barriers?

Responses #27:

{

The previous Figure 5c (now revised Fig. 6c) shows the dimer rupture (indicated by bead height increase step due to release of linker) and dimer re-formation (indicated by bead height decrease step due to looping of the linker) at low forces of 2-3.4 pN. The results demonstrate that the dimer formed by SR2-SR3 domains were weak, associated with lifetime of seconds within this force range.

The data can be used to estimate the force-dependent rupture rate and force-dependent dimerization rate of the designed dimer, from which the energy barriers could be estimated. Here we note that, the force-dependent rupture rate is independent of the long flexible linker. On the other hand, the force-dependent re-dimerization rate is related to the length of linker (at a same force, the longer the linker, the slower the re-dimerization rate).

In this study, we mainly focus on the rupture rate of the dimer, hence we analyzed the rupture rates of the dimer based on data presented in previous Fig. 5c, and included as a new figure panels in revised Figure 6.

Change List #20.

Also see in the Main Text: Page 23, Figure 6 and its captions.

}

Comments #28:

Would have been interesting to investigate the nature of the “multivalent interaction” mechanism. Maybe some SMD simulations, paired with control experiments that could disrupt this interaction, would be a good path to follow.

Responses #28:

{

The reviewer raises a very interesting and important point regarding the nature of “multivalent interaction” mechanism, and investigating the nature by combining with SMD simulations.

Since we are not specialized with SMD simulations, we are now looking for new collaboration with experts in the field of SMD simulation to further provide new insights with the molecular nature of the force-dependent rupture of the dimer. We also hope that the publication of the current study will raise the interests of experts of SMD field as well as other field to work together in the near future.

}

Comments #29:

Figure S8 shows a rather inhomogeneous distribution in the measured binding-unbinding kinetics, which the authors attribute to “weaker” and “strong” states. While such distributions are interesting and have been seen before for other proteins, this behavior does not seem to be addressed in the manuscript, except for “clearly distinguishable clusters”. What could be the mechanism for these clusters?

Responses #29:

{

The reviewer pointed out an interesting aspect of the un-dimerization behaviour of the α -actinin SR domains. In Figure S8, we monitored the dimerization and un-dimerization dynamics of the 3 SR dimer (SR2-SR3-SR4:SR1-SR2-SR3) at a low force of ~ 2.5 pN for over 60000 seconds. The dimer shows at least three distinct lifetime distributions in a cluster-like manner. We labelled the clusters as weak group (lifetimes in the scale of 100 seconds), weaker group (lifetimes in the scale of 10 seconds) and the strong group (lifetimes much great than 1000 seconds). We reason that such cluster-like distinct lifetime groups might be related to the different intermediate conformational states of the domain pairs in the dimer.

The structural studies of the α -actinin dimer have revealed that the anti-parallel SRs in the rod are organized with certain degree of re-orientation along the rod axis, resulting in twisting about 90° from one end to the other end. This suggest that the SR pairs should be not only paired, but also re-oriented with certain degree in the native dimerization state. The SR rod monomer contains four SR domains linked by a short helical linker. The linker between two neighbouring SR domains is a short rigid helix extended from the C-terminal helix of the former SR or the N-terminal helix of the later SR (Ref. 15, 21). When the SR rods form dimers, the SR domains in the rod not only need to be paired, but also need to be re-oriented to suit into the native conformation. Due to the rigidity of the helical linker and the domains, the domain-pairs may be trapped in certain intermediate conformational state where some of the domains pair and reorient into the native states, while some of the domains pair in non-native states, leading to the clustered distinct mechanical stability group. On the other hand, the domain-pair re-orientation also implies that natively paired and re-oriented domain-pairs may facilitate their neighbouring domain-pair’s re-orientation, leading to cooperativity of pairing and re-orientation of the domains, which in turn enhances the mechanical stability of the dimer. In the 3 SR domain dimer, such cooperativity effect may be weakened, leading to certain degree of freedom of the domain-pair’s re-orientation state, which results in multiple states of the dimer associated with distinct mechanical lifetimes. For instance, two of the SR pairs in the native orientation, while another pair stays in un-native pairing conformation. Consistently, in the 2-SR dimer, such cooperativity effect on the re-orientation further weakened, resulting in one weak mechanical lifetime group observed experimentally.

We thank the reviewer for raising up the simulating and constructive comments. We have included new discussions on the potential mechanisms in the revised manuscript.

Change List #11, 12 and 22.

Also see in the Main Text: Page 10, Paragraphs 3-5 & Page 11, Paragraphs 1-2; Page 25, Figure 8 and its captions.

}

Comments #30:

Other comments:

It might be useful to include a representation from the crystal structure determined in ref. 14 as a panel in Figure 1 and reference it during the Introduction, where this protein is described.

Responses #30:

{

Thanks for the reviewer's suggestion. We have revised the Fig. 1 and introduction accordingly.

Change List #15.

}

Comments #31:

It is not clear why the ancestral mutant brings to the table? How was it obtained? What sequence homogeneity does it have with the human actinin?

Responses #31:

{

The modern α -actinin family proteins are highly conserved in animals. They all contains four SR domains which are responsible for anti-parallel dimerization. The phylogenetic analysis of the α -actinin rod domain suggests that modern vertebrate α -actinins have evolved from ancestral forms containing one or two SR domains through intragenic duplication (Refs. 58,59). We therefore wondering how these ancestral rod dimers may be regulated by mechanical forces. A better understanding of the ancestral actinins may provide a better insight into the evolution of this superfamily. Hence, we choose the ancestral α -actinin like protein from *Entamoeba histolytica*, which is considered one of the most primitive protozoa. The *Entamoeba histolytica* α -actinin contains only two SRs in the rod, representing a critical node during the actinin evolution.

The DNA fragment containing the *Entamoeba histolytica* α -actinin SR domains gene was synthesized by IDT, based on the sequences from a previously published structural study on the protein (Ref. 60). The DNA fragment then was sub-cloned into our designed plasmid vector for single molecule stretching experiments.

Based on phylogenetic analysis of the α -actinin rod domain, the *Entamoeba histolytica* α -actinin SR1 and SR2 are most related to SR1 and SR4 of the modern α -actinin, while their sequence identities were low (~20% between the *Entamoeba histolytica* α -actinin SR1 and SR2 and the human α -actinin SR1 and SR4).

}

Comments #32:

Text in figures is too small to be legible. Because figures were not rasterized, using Acrobat to read the pdf file is laggy.

Responses #32:

{

Thanks for the reviewer's reminder. We have enlarged the texts in the figures accordingly.

Change List #15-22.

}

Comments #33:

Reviewer #3 (Remarks to the Author):

Overview: In this manuscript, the authors investigated mechanical stability of chimeric dimers of the cytoskeletal protein α -actinin isoforms. Dimers of α -actinin act in crosslinking actin to allow for cytoskeletal load bearing and assist in cellular mechanosensing and mechanotransduction. However, the mechanical stability of α -actinin dimers is not fully understood yet. They measured the effects of shear and unzip forces on α -actinin dimers. Magnetic tweezers were used to observed how force-induced rupturing of homo- and hetero-dimers of α -actinin affects their mechanical stabilities and lifetimes.

Overall, the experimental design to apply forces with magnetic beads was well thought out and the results could have significance in the field of cytoskeleton. Yet, the explanations of the main results and discussion needs to be clarified further to improve the manuscript.

Responses #33:

{

We thank the reviewer's supportive and constructive suggestions and comments. We have revised the manuscript accordingly.

}

Comments #34:

Major Comments:

1. How do the forces used to load the dimers relate in comparison to physiological forces the actin network may encounter?

Responses #34:

{

The reviewer raises an important question regarding the physiological forces load to the dimers. We discuss with three aspects: the physiological force range, the physiological force loading rate, the physiological force geometry.

Firstly, the primary source of forces applied to actin network and the actin associated proteins is the myosin fiber contraction. One myosin head can generate forces of around 3 pN (Ref. 39). A myosin fiber often contains 10-12 heads, which suggest a myosin fiber may generate forces of 3-36 pN, depending on how many heads bind to the actin (Ref. 40). Consistently, using FRET based cellular tension sensors, physiological forces experienced by many mechanosensitive proteins that linked to actin network have been estimated to be in the range of a few pN to tens of pN, such as talin, vinculin, α -catenin, cadherin, integrin, etc (Refs. 24-38). Stack of multiple myosin fibers could generate higher forces on the actin filament. The

higher forces on the actin filament could also be shared by multiple actinin dimers and other crosslinking proteins, hence, forces on individual protein/protein complex could still be reasonably estimated to be a few to tens of pN. Importantly, the forces experienced by the crosslinking proteins and associated proteins also depend on the mechanical stability of protein-protein interfaces of the force-bearing protein linkage. For instance, Dunn Lab has previously showed that the vinculin tail—actin interfaces could withstand forces up to about 10 pN, and the talin ABS3—actin interfaces could withstand forces up to about 20 pN (Refs. 48-49). We have previously showed that talin—vinculin interface could withstand forces up to 30 pN (Ref. 50). Altogether, forces of a few to tens of pN are reasonably physiological scale on individual mechanosensitive proteins.

Secondly, the force loading rates on individual mechanosensitive proteins could be roughly estimated by considering the dynamic interaction time scale. Previous FRAP experiments have estimated the time scale of many actin network related proteins are a few seconds to a few hundreds of seconds (Refs. 24,31,41). Hence, a rough estimation of the force loading rate could be in the order of pN/s. Consistently, a recent study by Ha Lab estimated the integrin loading rate in the range of from 0.5 pN/s to 4 pN/s, using a new FRET-based DNA force-loading sensor (Ref. 42).

Lastly, how the forces applied to the actinin dimer in vivo is also critical. Based on classic picture, the forces applied to the actinin dimer via its two ABDs. This leads to a shear-force geometry on the dimerized rod. In addition to this classic picture, there are also evidences suggest that the C-terminal EF hands of the actinin may also be stretched due to indirect association with actin network via adaptor proteins (Refs. 22-23). In this scenario, the force could also be applied to the actinin dimer via a N-terminal ABD and a C-terminal EF hands, leading to an unzip-force geometry. Hence, we studied the mechanical stability of the dimer under both the shear-force geometry and unzip-force geometry.

Altogether, in this study, we investigated the mechanical stability of the α -actinin dimer under physiologically relevant force-geometry within the physiologically relevant force range and force loading rate.

We thank the reviewer for raising this important question, and we have added corresponding introductions and discussions in the revised manuscript.

Change List #4.

Also see in the Main Text: Page 2, Paragraph 4 & Page 3, Paragraphs 1-2.

}

Comments #35:

2. For the force jump-cycle procedure, how do the authors know the dimer was refolded properly after the dimer has been fully unfolded and then allowed to refold? Do the different dimers always refold the same way?

Responses #35:

{

For the force jump-cycle procedure, as illustrated in the revised Fig. 2a, after fully unfolding of the domains at a target force, the tether was rapidly decreased to ~ 8 pN and followed by a slow force-decrease scan to ~ 1 pN with a loading rate of -0.1 pN s^{-1} , instead of directly jumped to ~ 1 pN forces. By the slow force-decrease scan from 8 pN to 1 pN, we clearly observed the refolding signals of the four SR domains and a re-dimerization signal of the fully folded SR domains. The four SR domains have characteristic refolding signals at forces of ~ 6 pN to ~ 4 pN at this loading rate, and the SR rods have characteristic re-dimerization signal at ~ 3 pN to

~ 2 pN. We therefore determine the proper refolding and re-dimerization of the dimer, based on the characteristic signals of the domain refolding and rods re-dimerization.

In most of the experiments of the force jump-cycle procedures (as well as the force-loading cycle procedures), the dimers refold and re-dimerized in the same manner. Occasionally, during the slow force-decrease scan, after three SR domains folded, the re-dimerization occurs almost concurrently with the last refolding event. It does not affect the mechanical stability of the dimer, evidenced by the rupture force (force-loading procedure) or lifetime (force jump cycle procedure) in the next scan.

We have included a typical time—bead height curves during multiple force jump-cycle procedure as the new figure 2a to improve the readability of the experimental procedure.

Change List #9, 15 and 16.

}

Comments #36:

3. It would be helpful to clearly explain how unzip-force stretching is distinct from shear-force stretching. Is the bead attached to the n what direction is the bead moving to induce an unzip force that differs from a shear force?

Responses #36:

{

Thanks for the reviewer's suggestion. Since the two rods in the dimer is anti-parallel, the force applied at the two N-termini is in a shear-force geometry, mimicking the physiological stretching geometry where the dimer is stretched via its two ABDs at N-termini. The force applied at a N-terminus of one rod and at the C-terminus of the other rod would lead to unzip-force geometry, mimicking the potential physiological stretching geometry where the C-terminus EF hands is also stretched. Hence, the key to design different force-geometries is designing the force-applying position on the dimer, which is achieved by designing the position of specific tethering tags on the protein.

To achieve shear-force stretching geometry in our experiments, as shown in the revised Fig. 1a, we designed a chimeric protein construct, containing (from N-terminus to C-terminus): a SpyTag, an α -actinin SR rod, a long flexible linker, a biotinylated AviTag, and another α -actinin SR rod. In experiments, the chimeric construct was specifically tethered to a SpyCatcher-coated coverslip surface via SpyTag, and to a streptavidin-coated paramagnetic bead surface via biotinylated AviTag. The two force-attaching points of the construct are SpyTag at the N-terminus of the first α -actinin SR rod and the biotinylated AviTag at the N-terminus of the second α -actinin SR rod. In this way, the two N-termini of the two rods are stretched in a shear force geometry.

To achieve unzip-force stretching geometry, as shown in the revise Fig. 3a, we designed another chimeric protein construct, in which the AviTag is placed prior the N-terminus of the first SR rod, while the SpyTag is placed after the C-terminus of the second SR rod. When the molecule is tethered between the bead and coverslip surface through these two tags, the N-terminus of the first rod and the C-terminus of the second rod are stretched unzip-force geometry.

We have revised the illustrations in revised Fig. 1 and Fig 3, and included more descriptions in the corresponding results sections, to enhance the readability of the experimental designs.

Change List #5, 10, 15 and 17

}

Comments #37:

4. How well do these SR based construct replicate the mechanical properties of the full-length α -actinin protein? Do the other domains known to be found in α -actinin not contribute to the mechanical strength?

Responses #37:

{

In current view, for a full-length α -actinin dimer, the N-terminal ABD domains are responsible for binding to F-actin. The SR domains are responsible for the dimerization of two monomers. The C-terminal EF-hands may regulate the binding of the ABD to F-actin in calcium-sensitive isoforms (α -actinin 1 and 4). In addition, both the SR domains and the EF-hands domain have been reported to be the binding partners of signalling proteins or adaptor proteins. In a word, once a full-length α -actinin dimer is activated under force by binding to F-actin via ABD, the mechanical stability of the dimer is dependent on the mechanical stability of the dimerized SR domains. The binding of signalling proteins or adaptor proteins to SRs may regulate the mechanical stability of the dimer. How the mechanical stability of the SR dimer may be regulated by the interactions with signalling proteins, as well as how such force-dependent interactions may affect the signalling will be interesting aspects to be investigated in future studies. We added a sentences in the discussion section to discuss the importance of studies on the binding factors of the SR dimers.

}

Comments #38:

5. The suggested kinetic modeling shown in Figure 7 is hard to grasp. Can the authors elaborate this suggested mechanism based on their results?

Responses #38:

{

In the previous Fig. 7, the model considers multivalent interaction kinetics of the dimer due to its multi-domain pairs. In the model, the dissociation transition path of the dimer is anticipated to initiate from the force-bearing ends under mechanical force. Since the dissociated domains remain in proximity due to the remaining of dimerized domain pairs, rapid re-association of the dissociated domain becomes possible, leading to stabilization through multivalency. Additionally, in the dissociated state, the stretched domain experiences force, potentially inducing unfolding and refolding of the force-bearing domain in a force-dependent manner. The unfolding of the domain hinders re-association. Consequently, the force-dependent lifetime of this protein--protein complex is governed by the force-dependent dissociation and re-association rates of the boundary domain pairs, the number of interacting domain pairs in the complex, as well as the force-dependent unfolding and refolding of the dissociated domains at the boundary. The inherent multivalency of the multiple domain dimer leads to a substantial increase in lifetimes. Here we note that, such a mechanical lifetime enhancement via multivalent interaction kinetics not only works for multi-domain complex, but also in general protein—protein complex with multiple interacting sites.

On the other hand, the kinetic model is still a simplified model compared to our experimentally investigated α -actinin dimers. The model did not consider several important parameters: the differences of the mechanical stability of individual SR domains, the differences of the mechanical stability of each SR pairs, and the effects of re-orientation of the domain pairs in the rod on its mechanical stability. Hence, the kinetic model results could not be fully compared with the experimental results. Nevertheless, the model highlights a key feature of the dimers, i.e, the dramatic strength enhancement by multiple domain interactions.

In the revised manuscript, enlightened by the reviewers' suggestions and comments, we included an additional molecular mechanism which possibly involves the re-orientations of the domain pairs in the dimer. This additional factor would explain the clustered dynamics of the dimer observed in experiments.

The previous Fig. 7 now is revised Fig. 8. Three new paragraphs are added in the discussion section to describe the models. The figure caption of the revised Fig. 8 is also updated correspondingly.

Change List #12 and 22.

}

Comments #39:

Minor Comments:

- The word "Multivalent" seems ambiguous.

Responses #39:

{

To avoid the potential ambiguity of the words, we rephrase the word to be "Multi-domain" in the paper.

Change List #1.

}

Comments #40:

- How similar is the sequence and structure of the *Entamoeba histolytica* α -actinin SR1 and SR2 compared to human α -actinin SRs?

Responses #40:

{

Based on phylogenetic analysis of the α -actinin rod domain (Refs. 58-59), the *Entamoeba histolytica* α -actinin SR1 and SR2 are most related to SR1 and SR4 of the modern α -actinin, while their sequence identities were low (~20% between the *Entamoeba histolytica* α -actinin SR1 and SR2 and the human α -actinin SR1 and SR4).

Structurally, all these SRs share characterized three alpha-helical bundle structure of spectrin repeat domains. Both the *Entamoeba histolytica* α -actinin and human α -actinin form anti-parallel dimers of SRs (Ref. 60). Interestingly, the human α -actinin dimer twists about 90° from one end to another end (via four SR pairs), while the *Entamoeba histolytica* α -actinin also twists about 90° from one end to another end (via two SR pairs) (Ref. 60). It suggests that the *Entamoeba histolytica* α -actinin dimer re-oriented more extensively than human α -actinin dimer. The differences in the domain pair number and the domain-pair re-orientation degrees may explain the differences in the mechanical lifetimes of the α -actinin dimers in *Entamoeba histolytica* and human.

We have included this point in the revised discussion section (Changes in Main Text: Page 11, Paragraph 2).

}

Comments #41:

- Figure 1 caption's title should include if the dimer was hetero or homo as was written for Figures 2-5.

Responses #41:

{

Thanks for the reviewer's reminder, we have added "homo" or "hetero" to the figure titles and captions accordingly.

Changes throughout the manuscript and figure captions.

}

Comments #42:

- The authors mentioned their measurements being within a physiological force range. Are there any references to show what forces the α -actinin experience in the cell?

Responses #42:

{

The reviewer raises an important question regarding the physiological force ranges and the forces the α -actinin experience in the cell. We discuss with three aspects: the physiological force range, the physiological force loading rate, the physiological force geometry.

Firstly, the primary source of forces applied to actin network and the actin associated proteins is the myosin fiber contraction. One myosin head can generate forces of around 3 pN (Ref. 39). A myosin fiber often contains 10-12 heads, which suggest a myosin fiber may generate forces of 3-36 pN, depending on how many heads bind to the actin (Ref. 40). Consistently, using FRET based cellular tension sensor, physiological forces experienced by many mechanosensitive proteins that linked to actin network have been estimated to be in the range of a few pN to tens of pN, such as talin, vinculin, alpha-catenin, cadherin, integrin, etc (Refs. 24-38). Stack of multiple myosin fibers could generate higher forces on the actin filament. The higher forces on the actin filament could also be shared by multiple actinin dimers and other crosslinking proteins, hence, forces on individual protein/protein complex could still be reasonably estimated to be a few to tens of pN. Importantly, the forces experienced by the crosslinking proteins and associated proteins also depend on the mechanical stability of protein-protein interfaces of the force-bearing protein linkage. For instance, Dunn Lab has previously showed that the vinculin tail—actin interfaces could withstand forces up to about 10 pN, and the talin—actin interfaces could withstand forces up to about 20 pN (Ref. 48-49). We have previously showed that talin—vinculin interface could withstand forces up to 30 pN (Ref. 50). Altogether, while there is no direct estimation of the forces experienced by actinin dimer, forces of a few to tens of pN are reasonably physiological scale on individual mechanosensitive proteins.

Secondly, the force loading rates on individual mechanosensitive proteins could be roughly estimated by considering the dynamic interaction time scale. Previous FRAP experiments have estimated the time scale of many actin network related proteins are a few seconds to a few hundreds of seconds (Refs. 24,31,41). Hence, a rough estimation of the force loading rate could be 0.1 pN/s to 10 pN/s. Consistently, a recent study by Ha Lab estimated the integrin loading rate ranged from 0.5 pN/s to 4 pN/s, using a new FRET-based DNA force-loading sensor (Ref. 42).

Lastly, how the forces applied to the actinin dimer in vivo is also critical. Based on classic picture, the forces applied to the actinin dimer via its two ABDs. This leads to a shear-force geometry on the dimerized rod. In addition to this classic picture, there are also evidence suggest that the C-terminal EF hands of the actinin may also be stretched due to indirect association with actin network via adaptor proteins. In this scenario, the force could also be applied to the actinin dimer via a N-terminal ABD and a C-terminal EF hands, leading to an

unzip-force geometry. Hence, we studied the mechanical stability of the dimer under both the shear-force geometry and unzip-force geometry.

Altogether, in this study, we investigated the mechanical stability of the actinin dimer under physiologically relevant force-geometry within the physiologically relevant force range and force loading rate.

We thank the reviewer for raising this important question, and we have added corresponding discussions in the revised manuscript.

Change List #4.

Also see in the Main Text: Page 2, Paragraph 4 & Page 3, Paragraphs 1-2.

}

REVIEWERS' COMMENTS

Reviewer #1 (Remarks to the Author):

The authors have addressed all of this reviewer's concerns.

Reviewer #3 (Remarks to the Author):

The authors have clearly addressed all my comments and questions in the revised manuscript as well as their responses. Thank you.